# Distinct architectural requirements for the *parS* centromeric sequence of the pSM19035 plasmid partition machinery

**Andrea Volante[1], Juan Carlos Alonso[2], Kiyoshi Mizuuchi[1]***

[1]Laboratory of Molecular Biology, National Institute of Diabetes and Digestive and Kidney Diseases, National Institutes of Health, Bethesda, United States; [2]Departamento de Biotecnología Microbiana, Centro Nacional de Biotecnología, Consejo Superior de Investigaciones Científicas, Madrid, Spain

**Abstract** Three-component ParABS partition systems ensure stable inheritance of many bacterial chromosomes and low-copy-number plasmids. ParA localizes to the nucleoid through its ATP-dependent nonspecific DNA-binding activity, whereas centromere-like *parS*-DNA and ParB form partition complexes that activate ParA-ATPase to drive the system dynamics. The essential *parS* sequence arrangements vary among ParABS systems, reflecting the architectural diversity of their partition complexes. Here, we focus on the pSM19035 plasmid partition system that uses a ParB$_{pSM}$ of the ribbon-helix-helix (RHH) family. We show that *parS*pSM with four or more contiguous ParB$_{pSM}$-binding sequence repeats is required to assemble a stable ParA$_{pSM}$-ParB$_{pSM}$ complex and efficiently activate the ParA$_{pSM}$-ATPase, stimulating complex disassembly. Disruption of the contiguity of the *parS*pSM sequence array destabilizes the ParA$_{pSM}$-ParB$_{pSM}$ complex and prevents efficient ATPase activation. Our findings reveal the unique architecture of the pSM19035 partition complex and how it interacts with nucleoid-bound ParA$_{pSM}$-ATP.

**\*For correspondence:**
kiyoshimi@niddk.nih.gov

**Competing interest:** The authors declare that no competing interests exist.

## Editor's evaluation

The work by Volante et al. studied a plasmid partition system, in which the authors discovered that four or more contiguous ParS sequence repeats are required to assemble a stable partitioning ParAB complex and activate the ParA ATPase. The work reveals a plasmid partitioning mechanism in which the mechanic property of DNA and its interaction with the partition complex may drive the directional movement of the plasmid.

## Introduction

Faithful chromosome segregation is essential for the proliferation of bacterial cells, and low-copy-number plasmids also need a robust partition mechanism for their stable inheritance. However, prokaryotes do not possess the mitotic machinery of eukaryotes: instead, alternative active DNA partition systems have evolved, among which ParABS systems (also called class I partition systems) are the most widespread. Basic ParABS systems consist of three components, a partition ATPase (ParA), a 'centromere'-binding protein (ParB), and a *cis*-acting centromere-like DNA site (*parS*).

ATP-activated ParA dimers bind nonspecific DNA (nsDNA) and localize to the nucleoid in vivo (*Ebersbach and Gerdes, 2001*; *Ebersbach and Gerdes, 2004*; *Pratto et al., 2008*; *Ringgaard et al., 2009*). The *parS* sites, often composed of multiple tandem repeats of binding consensus sequences for ParBs, demark the DNA-cargos that are translocated and positioned into the two halves of the cell before cell division by recruiting ParB molecules to assemble partition complexes (PCs). ParB

proteins fall into two structurally unrelated groups, dimeric helix-turn-helix (HTH), and dimeric ribbon-helix-helix (RHH) DNA-binding proteins. HTH-ParBs have been shown to bind not only site-specifically to their cognate *parS* sequences but also to spread many kilobase pairs into the DNA neighboring the *parS* sites (*Murray et al., 2006*; *Graham et al., 2014*; *Soh et al., 2019*; *Rodionov et al., 1999*; *Lynch and Wang, 1995*; *Breier and Grossman, 2007*; *Jalal et al., 2020*; *Osorio-Valeriano et al., 2019*; *Sanchez et al., 2015*). Therefore, they form large PCs containing many ParB molecules bound to condensed DNA around *parS*, and ParB spreading activity is essential for their partition function (*Rodionov et al., 1999*; *Debaugny et al., 2018*). In contrast, RHH-ParB does not spread beyond *parS* site judged by the absence of ParB-mediated silencing of *parS*-proximal gene expression (J. C. Alonso, unpublished observation), unlike HTH-ParBs (*Lynch and Wang, 1995*; *Rodionov et al., 1999*), but like HTH-ParBs, they interact with their cognate ParA proteins via their N-terminus (*Radnedge et al., 1998*; *Figge et al., 2003*; *Barillà et al., 2007*). RHH-ParB proteins also control the expression of the proteins involved in the partition system and plasmid copy number control by binding *parS* sites, which overlaps promoters of their genes (*de la Hoz et al., 2000*).

ParA–ParB interaction leads to activation of the ParA-ATPase, which is most efficient in the presence of nsDNA and *parS* DNA (*Ah-Seng et al., 2009*; *Chu et al., 2019*; *Pratto et al., 2008*; *Taylor et al., 2021*) and leads to dissociation of ParA from nsDNA. Interaction dynamics between the nucleoid-bound ParA and ParB in the PC prior to ATP hydrolysis and ParA dissociation determine the dynamics of the PC relative to the nucleoid. The common results of most systems in vivo appear to be the establishment of equidistant distribution of two or more PCs along the nucleoid(s) so that at cell division each daughter cell inherits at least one copy of the plasmid DNA (*Sengupta et al., 2010*; *Ringgaard et al., 2009*; *Lioy et al., 2015*; *McLeod et al., 2017*).

With biochemical findings and observations from live-cell imaging approaches accumulating in the field, combined with experiments using reconstituted cell-free reaction systems, a diffusion-ratchet mechanism of the ParABS partition was proposed. Here, the driving force for the DNA-cargo motion is generated by a propagating nucleoid-bound ParA distribution gradient (*Vecchiarelli et al., 2010*; *Hwang et al., 2013*; *Vecchiarelli et al., 2013*; *Vecchiarelli et al., 2014*; *Sugawara and Kaneko, 2011*). Additional models related to the diffusion-ratchet mechanism have also been proposed based on high-resolution imaging observations (*Lim et al., 2014*; *Le Gall et al., 2016*). While findings supporting diffusion-ratchet-type models for ParABS systems accumulate, many molecular details required to put the model on quantitatively solid ground are lacking. Large number of HTH-ParB dimers load onto a PC by ParB 'spreading' to facilitate partitioning. Resulting high concentration of ParB at the PC assures near saturation ParB-binding to the local nucleoid-bound ParA molecules, leading to efficient sensing of the local ParA distribution gradient by the PC. Combined with relatively short lifetime of the ParA/ParB-mediated cargo-nucleoid bridges and sufficient ParA-ATP reloading rate to the nucleoid would maintain enough cargo-nucleoid bridges needed to significantly suppress thermal diffusion of the cargo without completely blocking the motion of the cargo. Proper balance among these parameters is needed for efficient plasmid partition by diffusion-ratchet mechanism. Insufficient free diffusion suppression results in random diffusion of the cargo, and oversuppression by too many or too stable bridges blocks the cargo motion altogether (*Hu et al., 2015*; *Hu et al., 2017*; *Taylor et al., 2021*). However, it is unclear how a system with RHH-ParBs, which cannot spread ParB beyond the *parS* sites and thus can load only limited number of ParB dimers to a PC, can fit into the diffusion-ratchet paradigm. Nevertheless, live-cell imaging studies of systems that use RHH-ParB proteins showed an oscillating dynamic ParA distribution pattern on the nucleoid and PC chasing the receding tail of the ParA distribution (*Ringgaard et al., 2009*; *Lioy et al., 2015*; *McLeod et al., 2017*) similar to observations with the F-plasmid partition system and other related systems involving HTH-ParB proteins (*Hatano and Niki, 2010*; *Schofield et al., 2010*) for which there is accumulating evidence supporting the diffusion-ratchet mechanism. Therefore, we suspect systems with RHH-ParBs also operate via a diffusion-ratchet-type mechanism. A better understanding of how the PCs are organized for this group and a quantitative understanding of the interaction dynamics of these PCs with nucleoid-bound ParA molecules are critical for advancing our mechanistic understanding of these systems.

In this study, we focused on the pSM19035 partition system of *Streptococcus pyogenes*. This plasmid harbors a ParABS system composed of ParA$_{pSM}$ (also called Delta), an RHH ParB$_{pSM}$ (also called Omega), and six *parS$_{pSM}$* sites, each comprising 7–10 consecutive non-palindromic 7-bp-long

sequence repeats (5′-WATCACW-3′, symbolized by →) that overlap the promoter regions of *copS*, δ (coding ParA$_{pSM}$) and ω (coding ParB$_{pSM}$) genes. Each ParB$_{pSM}$ dimer binds one copy of the 7 bp *parS$_{pSM}$* consensus sequence. However, the affinity for a single repeat is low, while two dimers bind with high affinity to two direct (→→) or inverted (→←) repeats forming dimers of dimers (*de la Hoz et al., 2004*; *Weihofen et al., 2006*; *Welfle et al., 2005*). Within the structures of these ParB$_{pSM}$-*parS$_{pSM}$* complexes, DNA does not show significant curvature, and although the protein dimers bound to each DNA sequence repeat slightly deviates from dyad symmetry due to the bound DNA sequence asymmetry, full-size *parS$_{pSM}$*-ParB$_{pSM}$ complexes could be modeled as nearly straight DNA wrapped by left-handed spiral arrangement of ParB$_{pSM}$ dimers (*Weihofen et al., 2006*). Atomic force microscopy images of the complex involving seven *parS$_{pSM}$* consensus sequence repeats supported its straight arrangement and the lack of spreading (*Pratto et al., 2009*). ParA$_{pSM}$, unlike most other ParAs, forms dimers in solution in the absence of ATP (*Pratto et al., 2008*). Like other ParAs, it also undergoes a conformational transition upon binding ATP that increases its affinity for nsDNA (*Soberón et al., 2011*; *Pratto et al., 2008*). In the ATP-bound form, ParA$_{pSM}$ has been shown to bind nsDNA forming limited size patches containing several ParA$_{pSM}$ dimers at random location, instead of individual dimers independently distributed on the nsDNA (*Pratto et al., 2009*). In the presence of *parS$_{pSM}$* DNA and ParB$_{pSM}$, several *parS$_{pSM}$*-ParB$_{pSM}$ mini-filaments and ParA$_{pSM}$-nsDNA patches appeared to bind together to form large protein-DNA complexes bridging multiple DNA molecules (*Pratto et al., 2008*; *Pratto et al., 2009*; *Soberón et al., 2011*; *Lioy et al., 2015*). Interactions among these components fueled by ATP hydrolysis are thought to drive dynamic oscillations of the nucleoid-bound ParA$_{pSM}$ in vivo, which resembles those observed for the TP228 ParABS system (*Lioy et al., 2015*; *McLeod et al., 2017*).

Despite accumulating information summarized above, how the observed inter-molecular interactions coordinate the in vivo system dynamics resulting in robust plasmid partitioning remains a mystery. To approach this puzzle, here we studied functional requirements of the *parS$_{pSM}$* sequence-structure necessary for ParB$_{pSM}$-mediated activation of the ParA$_{pSM}$ ATPase. We found a minimum of four contiguous repeats of the *parS$_{pSM}$* heptad consensus sequence without a gap is necessary for full activity. Kinetics of the nsDNA-bound *parS$_{pSM}$*-ParB$_{pSM}$-ParA$_{pSM}$ complex formation and disassembly indicated the presence of a complex multistep process involved in ATPase activation.

## Results

### ParA$_{pSM}$ ATPase is synergistically activated by nsDNA, ParB$_{pSM}$, and *parS$_{pSM}$*-DNA

To define the requirements for stimulation of the ParA$_{pSM}$ ATPase activity by ParB$_{pSM}$, the steady-state ParA$_{pSM}$ ATP turnover rate was measured with varying concentrations of ParB$_{pSM}$ in the presence or absence of different duplex DNA cofactors. The turnover rate of ParA$_{pSM}$ ATPase alone is very low at 37°C (0.9 ± 0.1 ATP/ParA-dimer/h, N = 3; *Figure 1A*; no DNA). In the presence of a saturating concentration of double-stranded DNA (40 µg/ml pBR322 plasmid DNA plus 23–38 µg/ml double-stranded oligonucleotide with or without *parS$_{pSM}$* sequence), to which ATP-ParA$_{pSM}$ dimers can bind to support ATPase activation by ParB$_{pSM}$ (see below), no significant rate change was observed (1.0 ± 0.1 h$^{-1}$, N = 46, *Figure 1A*, pool of all measurements at [ParB$_{pSM}$] = 0). Next, effects of the *parS$_{pSM}$*-DNA in addition to 40 µg/ml nsDNA and ParB$_{pSM}$ were examined. ParA$_{pSM}$ ATP hydrolysis was stimulated up to ~20-fold ($k_{cat}$ = 20.5 ± 2.9 h$^{-1}$, N = 7) in the presence of ParB$_{pSM}$ and oligonucleotide duplex DNA containing one of the native arrangements containing seven *parS$_{pSM}$* heptad-sequence-repeats (7R-*parS$_{pSM}$*, →→←→→←←) (*Figure 1A*). The stimulation approached saturation around 2 µM ParB$_{pSM}$ at varying ParA$_{pSM}$ concentrations (*Figure 1—figure supplement 1A*). In contrast, when the *parS$_{pSM}$* DNA fragment was replaced with one having a scrambled sequence, ParB$_{pSM}$ stimulated ParA$_{pSM}$ ATPase activity only to 2.4 ± 1.1 h$^{-1}$ even in the presence of 8 µM ParB$_{pSM}$ that should have allowed non-*parS$_{pSM}$* DNA binding (N = 3*) (*Figure 1A*, scram). Similarly, a low level of ParA$_{pSM}$ ATPase stimulation was observed with ParB$_{pSM}$$^{1-27}$ peptide, which lacked the DNA-binding and dimerization domains (*Figure 1—figure supplement 1B*). These results demonstrated that full ParA$_{pSM}$-ATPase stimulation by ParB$_{pSM}$ requires specific *parS$_{pSM}$* interactions. Even the low *parS$_{pSM}$*-independent ATPase stimulation was not detected in the absence of DNA (*Figure 1A*, no DNA). Similarly, the low-level ATPase stimulation by non-*parS$_{pSM}$*-binding ParB$_{pSM}$$^{1-27}$ peptide was not detected without DNA (*Figure 1—figure supplement 1B*). We conclude ParA$_{pSM}$-nsDNA binding is required for the ATPase activation

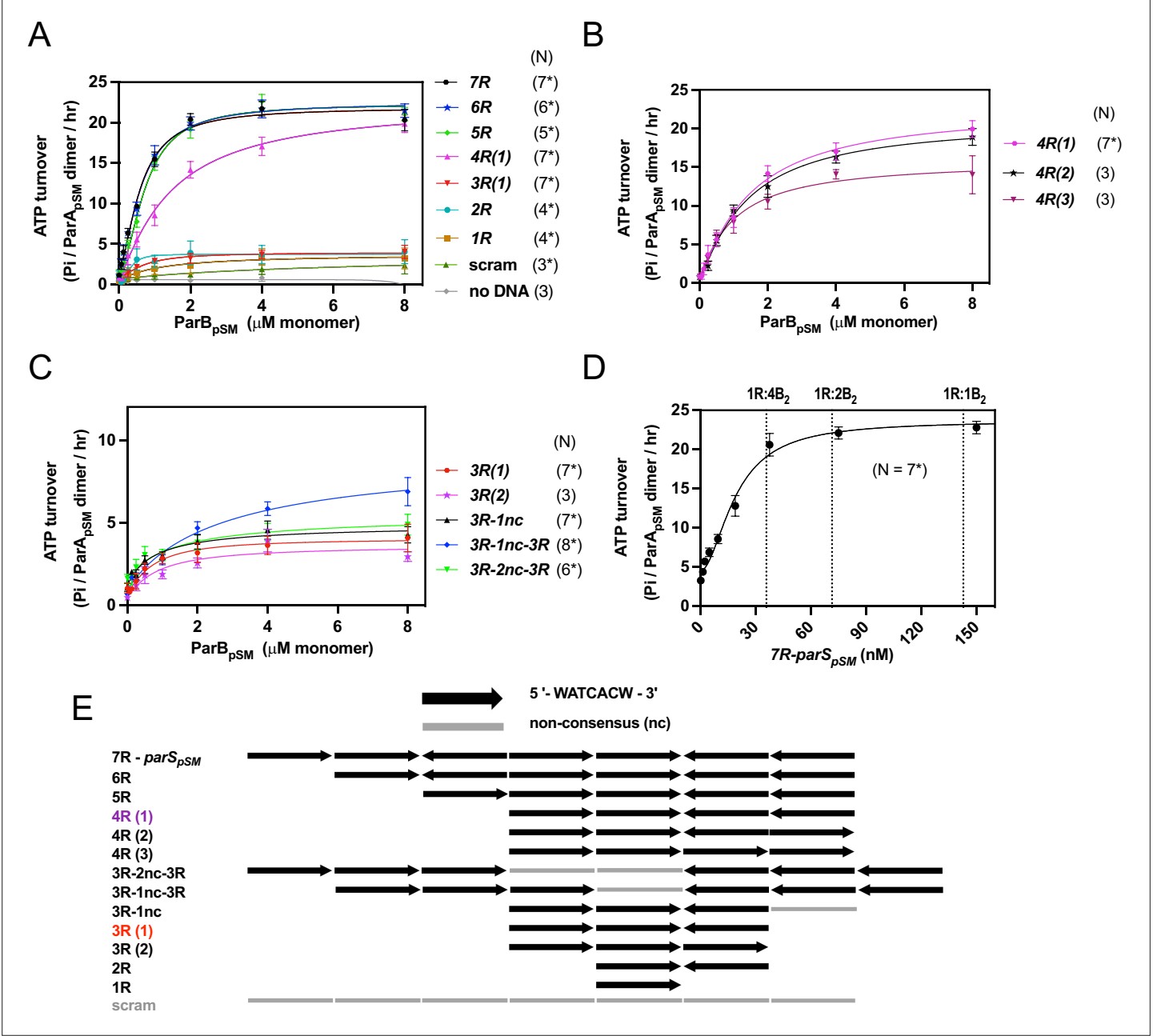

**Figure 1.** ParA$_{pSM}$ ATPase activation by ParB$_{pSM}$, parS$_{pSM}$, and nsDNA. (**A**) Efficient activation of ParA$_{pSM}$ ATPase by ParB$_{pSM}$ exhibits critical dependency on parS$_{pSM}$ heptad-sequence-repeat number. The ATPase reactions contained ParA$_{pSM}$ (2 µM), pBR322 DNA (60 µM in bp, unless noted otherwise), ParB$_{pSM}$ (at the concentration indicated), and parS$_{pSM}$ duplex substrates (4.4 µM of the 7 bp consensus sequence repeats) or equal amount of a scrambled sequence duplex (scram). (**B**) Comparison of parS$_{pSM}$ containing four contiguous repeats with different heptad orientation arrangements. (**C**) parS$_{pSM}$ containing different heptad arrangements of three contiguous repeats and two contiguous triple heptad repeats with a gap fails to fully activate the ParA$_{pSM}$ ATPase. (**D**) 7R-parS$_{pSM}$ concentration dependence of ParA$_{pSM}$ ATPase activity. Reaction mixtures contained ParA$_{pSM}$ (2 µM), ParB$_{pSM}$ (2 µM), pBR322 DNA (60 µM in bp), and increasing concentration of 7R-parS$_{pSM}$ duplex (duplex fragment concentration shown, the ratio of parS$_{pSM}$ heptad repeat sequence to ParB$_{pSM}$ dimers are indicated on top). (**E**) The numbers and arrangement of the heptad repeats of the parS$_{pSM}$ fragments used in this study (also see **Figure 1—figure supplement 2**). Data points represent means and standard errors of mean (SEM) of N repeated experiments (N* represents repeats for majority of data points, see **Figure 1—source data 1** for details). Curves were fitted after subtraction of the background in the absence of ParA$_{pSM}$ to an equation $v-v_0 = (v_{max}[B]^n)/(K_A^n + [B]^n)$. The maximum turnover rates ($v_{max}$) cited in the text represent mean ± 95% confidence intervals (symmetrized to the larger estimated errors from the mean for simplicity).

The online version of this article includes the following source data and figure supplement(s) for figure 1:

**Source data 1.**

*Figure 1 continued on next page*

Figure 1 continued

**Figure supplement 1.** ParA$_{pSM}$-ATPase stimulation by ParB$_{pSM}$ at different ParA$_{pSM}$ concentrations, ParB$_{pSM}$[1-27], and ParB$_{pSM}$-7R-parS$_{pSM}$ in the absence of nsDNA.

**Figure supplement 1—source data 1.**

**Figure supplement 2.** DNA duplex substrates.

**Figure supplement 2—source data 1.**

**Figure supplement 3.** Binding affinity of ParB$_{pSM}$ to 4R- and 3R-parS$_{pSM}$.

**Figure supplement 3—source data 1.**

by ParB$_{pSM}$, as reported for ParA$_F$ ATPase activation by ParB$_F$ (*Taylor et al., 2021*). Consistently, in the absence of nsDNA, excess ParB$_{pSM}$ relative to the concentration of 7R-parS$_{pSM}$ in the reaction inhibited ATPase stimulation, presumably competing with ParA$_{pSM}$ for parS$_{pSM}$ DNA binding (*Figure 1—figure supplement 1C*).

Formation of short clusters of ParA$_{pSM}$ molecules bound to nsDNA might be taken as a hint for cytoskeletal filament treadmilling models for this class of partition systems, rather than diffusion-ratchet model. However, the low maximum ATP turnover rate (~0.3 /min) observed would be too slow for a treadmilling filament model. Combined with low abundance of ParA$_{pSM}$ molecules inside a cell insufficient to form axial protein filaments, the treadmilling model is highly unlikely to fit this system.

## Efficient ParA$_{pSM}$-ATPase stimulation requires ParB$_{pSM}$ bound to parS$_{pSM}$-DNA with at least four contiguous heptad-sequence-repeats

We asked whether entire 7R-parS$_{pSM}$ is required for the full stimulation of ParA$_{pSM}$ ATPase by ParB$_{pSM}$. A series of deletions of parS$_{pSM}$ heptad-sequence-repeats were made and their ATPase stimulation activities were tested (*Figure 1E*). Efficient ATPase stimulation was observed when 6R (→←→→←←), 5R (→→→←←), or 4R (→→←←) was added along ParB$_{pSM}$ (*Figure 1A*). No significant difference was observed among 5R, 6R, or 7R; both half-saturation concentrations and the apparent $k_{cat}$ were comparable (*Figure 1A*). ParB$_{pSM}$-4R-parS$_{pSM}$ also induced similar rates of ParA$_{pSM}$ ATP hydrolysis, but a significantly higher concentration was required for full stimulation (*Figure 1A*). Different heptad orientation arrangements of 4R-parS$_{pSM}$ (→→←→ and →→→→) stimulated ParA$_{pSM}$ ATPase to a similar extent (*Figure 1B*). In contrast, 3R- (→→←), 2R- (→←), and 1R- (→) parS$_{pSM}$ were poor cofactors for ParB$_{pSM}$-dependent ATPase stimulation (apparent $k_{cat}$ = 3.2–3.6 ± 1.3–2.1 h$^{-1}$, N = 4–7*, *Figure 1A*), not significantly different from the scrambled sequence DNA. 3R-parS$_{pSM}$ fragments with different repeat arrangements behaved similarly to each other (*Figure 1C*). The ATPase stimulation by parS$_{pSM}$ DNA with 1–3 parS$_{pSM}$ repeats also appeared to saturate at around 2 μM ParB$_{pSM}$, indicating that the affinity of the nsDNA-bound ParA$_{pSM}$ to ParB$_{pSM}$ in the presence of truncated parS$_{pSM}$ was not limiting above ~2 μM ParB$_{pSM}$. Previously it has been shown that ParB$_{pSM}$ bound 3R, 4R DNA fragments of different sequence orientation combinations, or a 10R DNA fragment with roughly similar affinity that was >50-fold higher than nsDNA or 1R DNA (*de la Hoz et al., 2004*). We confirmed that ParB$_{pSM}$ binding was similarly strong for 4R and 3R duplex DNA ($K_D$ ~ 17 nM) and weak for nsDNA ($K_D$ ~ 1 μM, *Figure 1—figure supplement 3*). Therefore, the affinity of ParB$_{pSM}$ for the 3R-parS$_{pSM}$ DNA is not limiting the ParA$_{pSM}$ ATPase stimulation. In the above experiment, the length of parS$_{pSM}$ DNA fragments decreased as the number of parS$_{pSM}$ repeats decreased (*Figure 1E*, *Figure 1—figure supplement 2*). We tested longer 3R-parS$_{pSM}$ DNA fragments containing an additional non-parS$_{pSM}$ heptamer sequence (non-consensus, nc) (3R-1nc) and confirmed that parS$_{pSM}$-DNA fragment size was not a significant factor for the ATPase stimulation efficiency (*Figure 1C*).

These results suggested that ParA$_{pSM}$ ATP turnover is fine-tuned by the structural arrangement of the ParB$_{pSM}$ and parS$_{pSM}$ within the PC. ParB$_{pSM}$ assembles as a left-handed spiral to wrap parS$_{pSM}$ DNA without significantly distorting the DNA backbone geometry (*Weihofen et al., 2006*), implying that after approximately four repeats, ParB$_{pSM}$ dimers would make a full turn around parS$_{pSM}$, positioning themselves on the same face of the nucleoprotein filament. Thus, the position of the fourth repeats relative to that of the first repeat might be functionally important. To test this possibility, we examined the ParA$_{pSM}$ ATPase stimulation efficiency of parS$_{pSM}$ DNA fragments containing two copies of 3R sequences separated by 7 bp or 14 bp of non-consensus sequence (3R-1nc-3R and 3R-2nc-3R) (*Figure 1E*). The 3R-1nc-3R-parS$_{pSM}$ fragments showed only slightly higher stimulation ($k_{cat}$ = 8.3 ± 3.5

$h^{-1}$, N = 8*) compared to the single 3R-$parS_{pSM}$ fragment, and the 3R-2nc-3R was functionally indistinguishable from the single 3R-$parS_{pSM}$ fragment (*Figure 1C*). We conclude that disrupted 6R-$parS_{pSM}$ fragments are unable to recover the full stimulation of ParA$_{pSM}$ ATPase activity. These findings indicate that the ParB$_{pSM}$-ParA$_{pSM}$ interactions differ when ParB$_{pSM}$ dimers are bound to ≥4 contiguous heptad repeats compared to ParB$_{pSM}$ dimers unbound to the repeats or bound to fewer or a disrupted array of heptad repeats.

## Substoichiometric concentration of *parS$_{pSM}$* relative to ParB$_{pSM}$ is sufficient to fully activate ParA$_{pSM}$ ATPase

We originally assumed that for efficient ParA$_{pSM}$ ATPase activation by the ParB$_{pSM}$-$parS_{pSM}$ complex all ParB$_{pSM}$ dimers need to be bound to $parS_{pSM}$ and accordingly maintained stoichiometrically excess $parS_{pSM}$-sequence concentration relative to ParB$_{pSM}$-dimers in the reaction. To test this assumption, we next changed the concentration of the 7R-$parS_{pSM}$ DNA fragment while keeping the concentrations of ParA$_{pSM}$ and ParB$_{pSM}$ both at 2 µM. Contrary to our expectation, significantly lower heptad consensus sequence concentrations of the 7R-$parS_{pSM}$ DNA fragment compared to ParB$_{pSM}$ dimers (B$_2$) were sufficient for ATPase activation (*Figure 1D*). If the original assumption was correct, 2 µM ParB$_{pSM}$ should have required 1 µM consensus sequence repeats (143 nM 7R-$parS_{pSM}$) for full activation. According to the results of *Figure 1A*, at high 7R-$parS_{pSM}$ concentration, half-activation by ParB$_{pSM}$ required ~650 nM ParB$_{pSM}$, which would have required ~325 nM $parS_{pSM}$ consensus sequence if full binding was needed. Observed half-saturation $parS_{pSM}$ repeat concentration was ~130 nM (~19 nM 7R-$parS_{pSM}$) or less. Thus, assuming most of the ParB$_{pSM}$ molecules in our preparations are active, it appears that at any given time, less than half of the ParB$_{pSM}$ dimers need to be in complex with 7R-$parS_{pSM}$ to exert full ATPase activation.

## Stimulation of ParA$_{pSM}$ release from nsDNA-carpet by ParB$_{pSM}$ requires *parS$_{pSM}$* DNA with four or more heptad repeats

Next, we examined how the dissociation of ParA$_{pSM}$-ATP dimers from nsDNA is influenced by ParB$_{pSM}$ in the presence of $parS_{pSM}$ DNA. For these experiments, we used an nsDNA-carpeted two-inlet flow cell observed under TIRF microscopy as described in 'Materials and methods' (*Vecchiarelli et al., 2013*; *Figure 2A*). We used a ParA$_{pSM}$-GFP fusion protein and a ParB$_{pSM}$-*cys* conjugated to Alexa647 to facilitate visualization of protein association-dissociation to/from the nsDNA-carpet. Stimulation of ParA$_{pSM}$-GFP ATPase activity by ParB$_{pSM}$ and 7R-*parS* was comparable to wild-type ParA$_{pSM}$ (*Figure 2— figure supplement 1*). First, we tested the association/dissociation of individual proteins (bound at 1 µM) to/from the nsDNA-carpet. The affinity of ParB$_{pSM}$-Alexa647 (1% labeled) to nsDNA was weak. ParB$_{pSM}$-Alexa647 in the absence of ParA$_{pSM}$ accumulated on the DNA-carpet reaching a density of ~3000 ± 500 dimers/µm$^2$ after 15 min of constant flow (5 µl/min) (*Figure 2—figure supplement 2A*, bottom). When the sample solution was switched to a buffer without protein, most ParB$_{pSM}$ rapidly dissociated with kinetics that can be fitted to a double-exponential function ($k_{off\text{-}fast}$ = 5.1 ± 0.7 min$^{-1}$ [73%], $k_{off\text{-}slow}$ = 0.11 ± 0.04 min$^{-1}$, N = 3; all $k_{off}$s reported in this study represent apparent pseudo-first order rate constants) (*Figure 2—figure supplement 2B*, bottom). ParA$_{pSM}$-GFP preincubated with ATP and Mg$^{2+}$ reached a saturation density on the nsDNA-carpet of 4.25 (±0.06) × 10$^4$ dimers/µm$^2$ (N = 4*), whereas in the absence of ATP, less than 1000 dimers/µm$^2$ of ParA$_{pSM}$ bound to the nsDNA-carpet (*Figure 2—figure supplement 2A*, top). Next, 1 µM ParA$_{pSM}$-GFP preincubated with ATP was flowed onto the nsDNA-carpet until 5–10% of saturation density (~4000 dimers/µm$^2$), at which point the flow was switched to buffer containing ATP without ParA$_{pSM}$-GFP. A small fraction of ParA$_{pSM}$-GFP (less than ~5%) dissociated within ~1 min and the remainder dissociated slowly with $k_{off}$ of <0.013 min$^{-1}$ (N = 3, *Figure 2B*).

The addition of competitor nsDNA (71 nM scrambled 55 bp duplex DNA) in the wash buffer slightly increased the $k_{off}$ of majority (~93%) fraction of ParA$_{pSM}$-GFP ($k_{off}$ = 0.075 ± 0.001 min$^{-1}$, N = 3), perhaps in part by competing with rebinding of ParA$_{pSM}$-GFP-ATP to the nsDNA-carpet (*Figure 2B*). When 1 µM ParB$_{pSM}$ (unlabeled or 1% Alexa647 labeled) was added to the wash buffer without competitor nsDNA, $k_{off}$ of ParA$_{pSM}$-GFP from nsDNA-carpet reached 0.17 ± 0.01 min$^{-1}$ (N = 4, *Figure 2B*). ParA$_{pSM}$-GFP dissociation from the nsDNA-carpet was accelerated when wash solution contained 71 nM 7R-$parS_{pSM}$ DNA (0.5 µM of the consensus sequence repeat) and 1 µM ParB$_{pSM}$ ($k_{off}$ = 1.97 ± 0.09) after a brief period of initial slower dissociation rate (see below for details) (*Figure 2B*). This was ~12 or

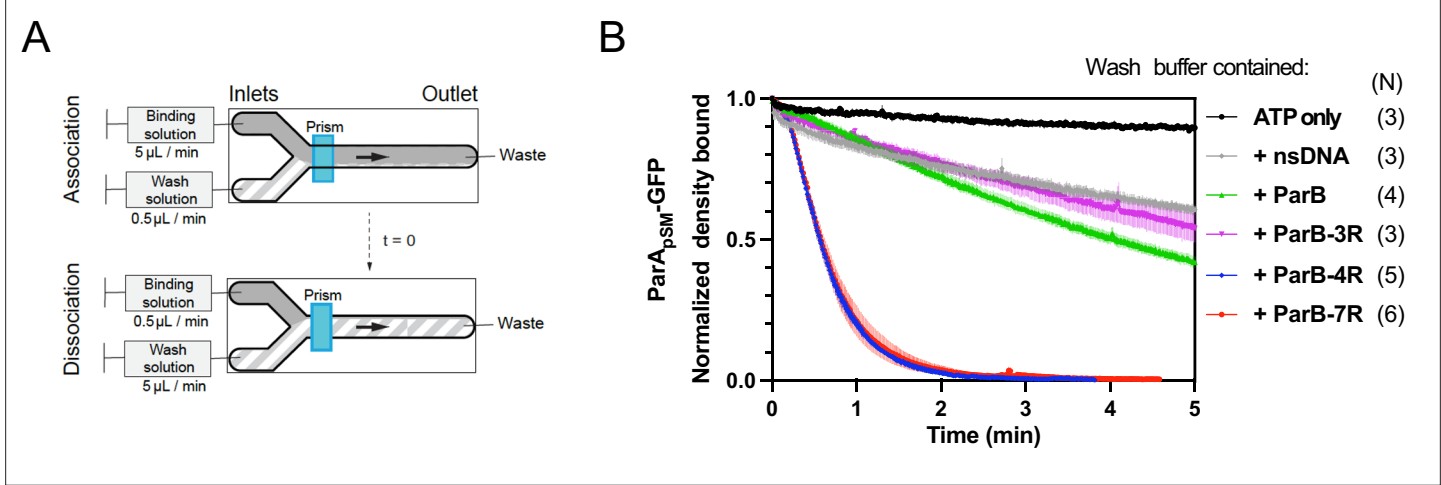

**Figure 2.** Kinetics of ParA$_{pSM}$ disassembly from the nsDNA-carpet. (**A**) Schematic of two-inlet flow cell used for visualizing the association and dissociation of fluorescent proteins on nsDNA-carpet. Binding and washing solutions each containing proteins, double-stranded DNA fragments or buffer alone as specified were infused at different flow rates (5 µl/min or 0.5 µl/min) from two inlets on the left into a Y-shaped flow cell. At the middle of the flow channel just downstream of the flow convergence point where the observations are made, content of the faster infusion syringe flows over the observation point. By switching the flow rates of the two syringes, protein binding to the nsDNA-carpet and dissociation during the washing cycle can be recorded. (**B**) ParB$_{pSM}$ in the presence of parS$_{pSM}$ with at least four contiguous ParB$_{pSM}$-binding sequence repeats stimulates the ParA$_{pSM}$-GFP dissociation from the nsDNA-carpet. ParA$_{pSM}$-GFP (1 µM) preincubated with 1 mM ATP was infused into the nsDNA-carpeted flow cell at 5 µl/min, while the washing solution containing the specified components was infused at 0.5 µl/min. When the ParA$_{pSM}$-GFP density on the nsDNA-carpet reached 5–10% of the saturation density (t = 0) (~4000 ParA$_{pSM}$ dimers/µm$^2$), the flow rates were switched to start the wash with solution containing: buffer alone, 55 bp scramDNA fragment (65 nM in bp), or with ParB$_{pSM}$ (1 µM) without or with different parS$_{pSM}$ fragment (0.5 µM parS$_{pSM}$ heptad repeat sequence). The Y-axis shows the ParA$_{pSM}$-GFP intensity normalized to that at t = 0. Each time point represents the mean with error bar corresponding to the standard errors of mean (SEM) of N repeated experiments.

The online version of this article includes the following source data and figure supplement(s) for figure 2:

**Source data 1.**

**Figure supplement 1.** ATPase activity of ParA$_{pSM}$-GFP and the hydrolysis-deficient ParA$_{pSM}^{D60E}$-GFP.

**Figure supplement 1—source data 1.**

**Figure supplement 2.** ParA$_{pSM}$, ParA$_{pSM}^{D60E}$, and ParB$_{pSM}$ interactions with the DNA-carpet measured each separately.

**Figure supplement 2—source data 1.**

**Figure supplement 3.** ParB$_{pSM}$–activated dissociation of ParA$_{pSM}$-GFP or ParA$_{pSM}^{D60E}$-GFP bound to nsDNA-carpet to saturation density.

**Figure supplement 3—source data 1.**

---

~26-fold higher $k_{off}$ compared to ParB$_{pSM}$ or nsDNA wash, respectively. Similarly, 4R-parS$_{pSM}$ DNA supported accelerated ParA$_{pSM}$-GFP release by ParB$_{pSM}$, but 3R-parS$_{pSM}$ DNA did not (***Figure 2B***). The results paralleled those shown in ***Figure 1A*** solidifying the notion that a functional parS$_{pSM}$ must carry four copies of ParB$_{pSM}$ dimer-binding sequence repeats.

The dependence of ParA$_{pSM}$ dissociation behaviors on the heptad repeat number of the parS$_{pSM}$ fragment combined with 1 µM ParB$_{pSM}$ in the wash solution when the wash was started with ParA$_{pSM}$ bound to the nsDNA-carpet at a saturating density was qualitatively similar. However, the dissociation kinetics were significantly slower (***Figure 2—figure supplement 3A***), likely reflecting the approximately tenfold higher starting ParA$_{pSM}$-GFP density on the nsDNA-carpet, for which the concentration of ParB$_{pSM}$ used likely was limiting.

## Accelerated release of ParA$_{pSM}$ from DNA-carpet starts only after accumulation of ParB$_{pSM}$ on the carpet in the presence of 7R- or 4R-parS$_{pSM}$

The accelerated ParA$_{pSM}$-GFP release from the nsDNA-carpet when washed with ParB$_{pSM}$-4R or –7R parS$_{pSM}$ complexes started after an initial slower rate of ParA$_{pSM}$ release (***Figure 2B***). This time lag prompted us to examine the binding kinetics of ParB$_{pSM}$ to the ParA$_{pSM}$-bound nsDNA-carpet at

different concentrations of ParB$_{pSM}$. Upon switching the flow to the washing solution without *parS$_{pSM}$*, ParB$_{pSM}$ bound to the ParA$_{pSM}$-bound nsDNA-carpet to a peak density of ~3000 dimers/µm$^2$ and then quickly decreased to a plateau density of ~2000 dimers/µm$^2$ (***Figure 3Ad***). The initial overshoot of ParB$_{pSM}$ binding appeared to roughly coincide with the fast dissociation phase of ParA$_{pSM}$. During the subsequent slow dissociation of ParA$_{pSM}$, ParB$_{pSM}$ density on the nsDNA-carpet remained relatively constant with the protein ratio reaching ~1 after several min of washing with 1 µM ParB$_{pSM}$ (***Figure 3Ad***).

In the presence of 1 µM ParB$_{pSM}$ and 7R-*parS$_{pSM}$* in the wash solution, ParA$_{pSM}$-GFP release from the nsDNA-carpet took place in two phases (***Figure 3Aa***). First, ParB$_{pSM}$ accumulated on the ParA$_{pSM}$-bound nsDNA-carpet to a density twofold or more in excess of the carpet-bound ParA$_{pSM}$. During this phase, ParA$_{pSM}$ dissociated from the nsDNA-carpet relatively slowly. Next, as the binding of ParB$_{pSM}$ stopped and began to quickly dissociate ($k_{off}$ = 2.4 ± 0.1 min$^{-1}$, N = 5), ParA$_{pSM}$ dissociation also accelerated to a higher rate ($k_{off}$ = 1.8 ± 0.05 min$^{-1}$, N = 5) (***Figure 3Aa***). Near-complete dissociation of both proteins from the nsDNA-carpet occurred within a few minutes. Biphasic dissociation kinetics of ParA$_{pSM}$ was clearer when the wash solution contained lower concentrations of the 7R-*parS$_{pSM}$*-ParB$_{pSM}$ complex, as ParB$_{pSM}$ accumulated on the ParA$_{pSM}$-bound nsDNA-carpet slower, and it took longer to transition to the accelerated dissociation phase (***Figure 3Ae,i***). This indicates that the initial nsDNA-carpet-bound ParA$_{pSM}$-ParB$_{pSM}$ complex is not activated for ParA$_{pSM}$ dissociation from nsDNA, and slow transition of the nsDNA-bound complex is necessary to start the ParA$_{pSM}$ disassembly.

We next examined the ability of 4R- and 3R-*parS$_{pSM}$* DNA to accelerate ParA$_{pSM}$-GFP dissociation in the presence of ParB$_{pSM}$. Switching 7R-*parS$_{pSM}$* in the wash solution to 4R-*parS$_{pSM}$* (1 µM ParB$_{pSM}$, 125 nM 4R DNA) yielded qualitatively similar two-step dissociation curves (***Figure 3Ab***). The $k_{off}$ of ParA$_{pSM}$ for the accelerated dissociation phase (1.9 ± 0.06 min$^{-1}$, N = 5) was roughly the same as in the presence of 7R-*parS$_{pSM}$*. After the peak of binding, ParB$_{pSM}$ dissociated from the nsDNA-carpet with $k_{off}$ of 3.9 ± 0.2 min$^{-1}$, N = 5. The dissociation burst of ParA$_{pSM}$-GFP was triggered in the presence of 4R-*parS$_{pSM}$* at ParB$_{pSM}$/ParA$_{pSM}$ molar ratio of ~1.7 compared to ~2.8 in the presence of 7R-*parS$_{pSM}$* (at 1 µM ParB$_{pSM}$). This difference likely in part reflects the fact that an individual ParB$_{pSM}$-*parS$_{pSM}$* complex with 4R-*parS$_{pSM}$* contains fewer ParB$_{pSM}$ dimers than the 7R-*parS$_{pSM}$* complex contains. Also, at a given ParB$_{pSM}$ concentration in the wash solution, washing with 7R-*parS$_{pSM}$* took roughly twice as long to reach the peak ParB$_{pSM}$/ParA$_{pSM}$ ratio to start the rapid ParA$_{pSM}$ dissociation phase compared to the 4R-*parS$_{pSM}$* wash (***Figure 3B***). We believe this likely reflects, in part, a lower concentration of the larger ParB$_{pSM}$-7R-*parS$_{pSM}$* complex than the complex with 4R-*parS$_{pSM}$*. The accelerated phase of ParA$_{pSM}$ $k_{off}$ appeared higher in the presence of higher concentrations of 7R- or 4R-*parS$_{pSM}$*-ParB$_{pSM}$ complex in the wash solution (***Table 1***).

Unlike the results above using 7R- or 4R-*parS$_{pSM}$* in the wash solution, washing experiments with 3R-*parS$_{pSM}$* (1 µM ParB$_{pSM}$, 166 nM 3R-*parS$_{pSM}$*) did not exhibit an accelerated ParA$_{pSM}$-GFP dissociation phase, as in the absence of *parS$_{pSM}$* (***Figure 3Ac,d***). ParB$_{pSM}$ quickly accumulated onto the ParA$_{pSM}$-bound nsDNA-carpet up to a ParB$_{pSM}$/ParA$_{pSM}$ ratio of ~1.3 (1 µM ParB$_{pSM}$). Thus, ParB$_{pSM}$ appears to be able to associate with nsDNA-bound ParA$_{pSM}$ to a stoichiometry, which is unlikely to be the factor preventing stimulation of ParA$_{pSM}$ dissociation. Both ParA$_{pSM}$-GFP and ParB$_{pSM}$ dissociated from their peak density on the carpet with double-exponential kinetics roughly in parallel (ParA$_{pSM}$-GFP; $k_{off\text{-}fast}$ = 3.8 min$^{-1}$ ± 0.4 min$^{-1}$ [~19%], $k_{off\text{-}slow}$ = 0.07 ± 0.003 min$^{-1}$: ParB$_{pSM}$; $k_{off\text{-}fast}$ = 1.4 ± 0.3 min$^{-1}$ [~24%], $k_{off\text{-}slow}$ = 0.05 ± 0.004 min$^{-1}$, N = 3) (***Figure 3Ac***).

## ATP hydrolysis triggers fast ParA$_{pSM}$ dissociation from the nsDNA-carpet

We next asked whether the accelerated dissociation of ParA$_{pSM}$ from the nsDNA-carpet in the presence of *parS$_{pSM}$* fragments depends on ATP hydrolysis by ParA$_{pSM}$. Among ParA$_{pSM}$ homologues, a conserved Asp in the Walker A' motif is critical for ATP hydrolysis (***Pratto et al., 2008***; ***Park et al., 2012***). We prepared the ParA$_{pSM}$$^{D60E}$-GFP fusion protein; it exhibited no significant residual ATPase activity and faster ATP-dependent nsDNA binding compared to the wild-type protein (***Figure 2—figure supplement 1***, ***Figure 2—figure supplement 2A***, top). The apparent $k_{off}$ of the majority fraction (>80%) of nsDNA-bound ParA$_{pSM}$$^{D60E}$-GFP washed with ATP-containing buffer (0.015 ± 0.007 min$^{-1}$, N = 2) was not significantly different from ParA$_{pSM}$-GFP (0.014 ± 0.002 min$^{-1}$, N = 2) (***Figure 2—figure supplement 2B***, top). However, when the wash solution contained 250 nM ParB$_{pSM}$ and 36 nM

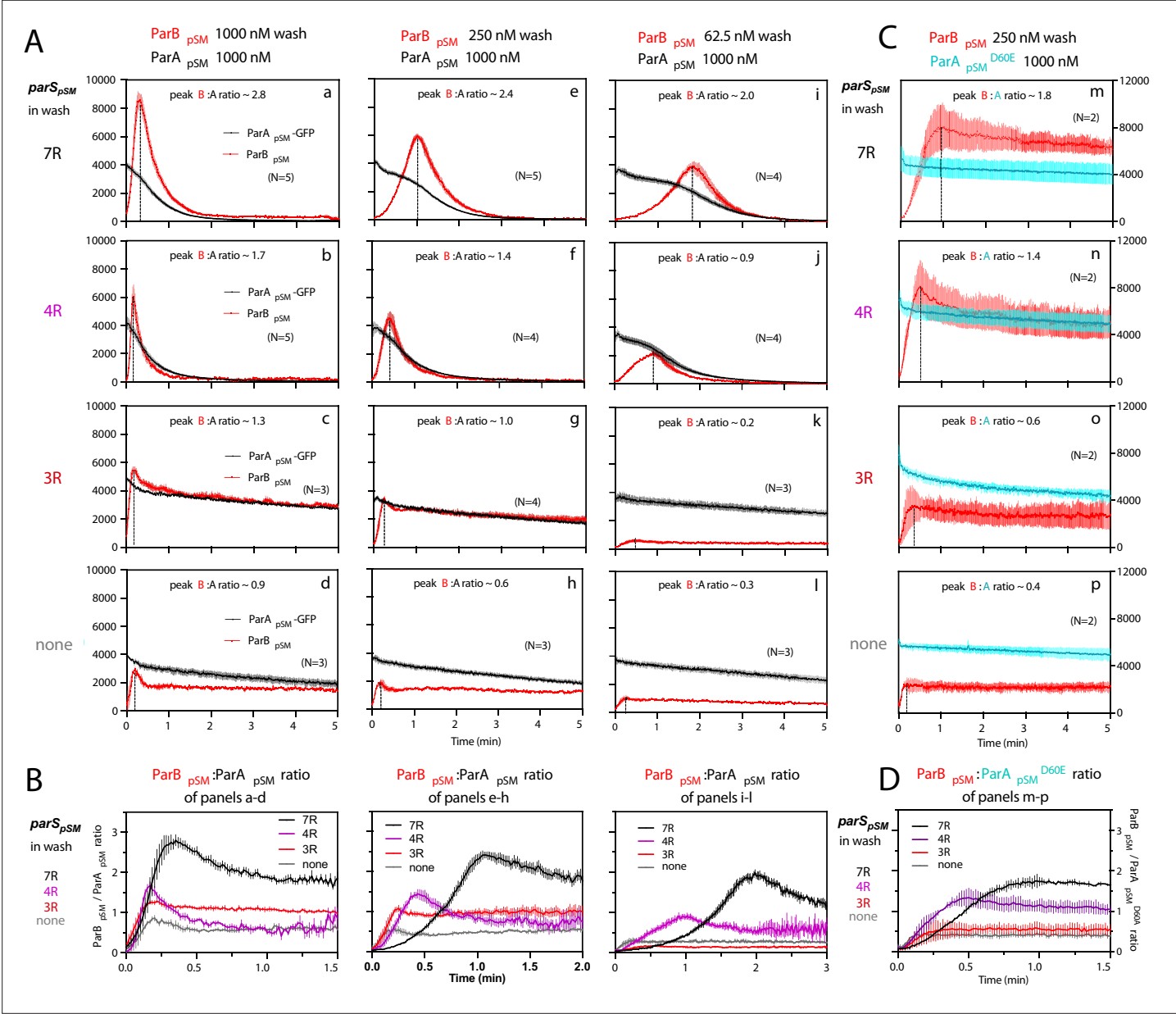

**Figure 3.** ParB$_{pSM}$-parS$_{pSM}$ concentration affects kinetics of ATP hydrolysis-dependent ParA$_{pSM}$ disassembly from the nsDNA-carpet. (**A**) ParA$_{pSM}$-GFP (1 μM) preincubated with 1 mM ATP was infused into the nsDNA-carpeted flow cell at 5 μl/min and when the ParA$_{pSM}$-GFP density on the nsDNA-carpet reached ~10% of the saturation density, the flow over the observation area was switched (t = 0) to wash solution containing ParB$_{pSM}$ ([**a–d**] 1000 nM, [**e–h**] 250 nM, or [**i–l**] 62.5 nM) and the stoichiometric concentration of parS$_{pSM}$ fragments indicated in the 'parS$_{pSM}$ in wash' column on the left side of each row of panels. Fluorescence signal was converted to protein density and plotted (dimers/μm², ParA$_{pSM}$-GFP: black; and ParB$_{pSM}$-Alexa647: red). Each time point represents mean and standard errors of mean (SEM) (vertical spread) of N repeated experiments. Dashed vertical lines indicate the peak ParB$_{pSM}$ density on the nsDNA-carpet. (**B**) The time course of the ParB$_{pSM}$:ParA$_{pSM}$ molar ratio (B/A) for the four panels in the columns in (**A**) with matching ParB$_{pSM}$ concentration with different parS$_{pSM}$ (7R: black; 4R: purple, 3R: red; none: gray) in the wash solution. (**C**) ATP hydrolysis is required for accelerated ParA$_{pSM}$ release from the nsDNA-carpet. The experiments shown in (**A**) (middle column) were repeated using ParA$_{pSM}$$^{D60E}$-GFP (1 μM) bound to the nsDNA-carpet and ParB$_{pSM}$ (250 nM) plus parS$_{pSM}$ fragment, (**m**) 7R, (**n**) 4R, (**o**) 3R, or (**p**) without parS$_{pSM}$ (125 nM heptad concentration) in the wash solution. Fluorescence signal was converted to protein density (dimers/μm², ParA$_{pSM}$$^{D60E}$-GFP: cyan; ParB$_{pSM}$-Alexa647: red) and plotted. Each time point represents mean and SEM (vertical spread) of N experiments. (**D**) The time course of the ParB$_{pSM}$:ParA$_{pSM}$$^{D60E}$ molar ratio (B/A$^{D60E}$) for the four panels of (**C**) (7R: black; 4R: purple; 3R: red; none: gray). Each time point represents the mean with error bar corresponding to the SEM of N repeated experiments.

The online version of this article includes the following source data for figure 3:

**Source data 1.**

**Table 1.** The accelerated ParA$_{pSM}$ dissociation phase of the curves in **Figure 3Aa,c,i and b,f,j** after the peak ParB$_{pSM}$/ParA$_{pSM}$ ratio points were fitted to single exponential curves to estimate the $k_{off}$ values.

The means with error bars corresponding to the standard errors of mean (SEM) of N repeated experiments in **Figure 3** are shown.

| [ParB$_{pSM}$] (nM) | 7R-parS$_{pSM}$ | 4R-parS$_{pSM}$ |
|---|---|---|
| | ParA$_{pSM}$ $k_{off}$ (min$^{-1}$) | ParA$_{pSM}$ $k_{off}$ (min$^{-1}$) |
| 1000 | 1.808 ± 0.045 | 1.909 ± 0.057 |
| 250 | 1.495 ± 0.024 | 1.462 ± 0.040 |
| 62.5 | 1.184 ± 0.041 | 1.101 ± 0.033 |

7R-parS$_{pSM}$, ParA$_{pSM}$$^{D60E}$-GFP dissociation from the nsDNA-carpet was only a factor of ~2 faster (**Figure 3Ca**; $k_{off}$ = 0.029 ± 0.005 min$^{-1}$, N = 2) than washing with ATP-buffer. As washing started, the ParB$_{pSM}$ associated with the carpet-bound ParA$_{pSM}$$^{D60E}$-GFP to a density well beyond that of ParA$_{pSM}$$^{D60E}$-GFP with similar association kinetics as with the carpet-bound ParA$_{pSM}$-GFP (**Figure 3Ae**). The ParB$_{pSM}$/ParA$_{pSM}$ ratio bound to the nsDNA-carpet at the initial ParB$_{pSM}$ overshoot peak was ~1.75 in the presence of 7R-parS$_{pSM}$, ~1.3 with 4R-parS$_{pSM}$, and ~0.6 with 3R-parS$_{pSM}$ (**Figure 3D**). After the peak density, ParB$_{pSM}$ dissociated from the nsDNA-carpet slowly ($k_{off}$ = 0.036 ± 0.003 min$^{-1}$, N = 2) (**Figure 3Ca**). Thus, the accelerated DNA dissociation of ParA$_{pSM}$ by the active ParB$_{pSM}$-parS$_{pSM}$ complex is coupled to ATP hydrolysis, and in the absence of hydrolysis, ParA$_{pSM}$$^{D60E}$-ATP dimers stayed stably on nsDNA while associated with active parS$_{pSM}$-ParB$_{pSM}$ complex.

## ParB$_{pSM}$ associates with nsDNA-bound ParA$_{pSM}$ more stably in the presence of 7R- or 4R-parS$_{pSM}$ prior to ATP hydrolysis

In the experiments of **Figures 2 and 3**, ParA$_{pSM}$-ATP dimers were bound to the nsDNA-carpet first without ParB$_{pSM}$, and during the wash phase of the experiments, the carpet-bound ParA$_{pSM}$ was constantly exposed to a solution containing ParB$_{pSM}$ and parS$_{pSM}$. Because ParB$_{pSM}$ and parS$_{pSM}$ dissociating from carpet-bound ParA$_{pSM}$ could be exchanged by those in the wash solution, the data did not report the stability of ParA$_{pSM}$-ParB$_{pSM}$ interaction on the nsDNA-carpet. In the next set of experiments, we preincubated an equimolar ratio of ParA$_{pSM}$-GFP (1 µM), ParB$_{pSM}$ (1 µM), and 7R-, 4R-, or 3R-parS$_{pSM}$ (0.5 µM total concentration of the consensus sequence repeats) in the presence of ATP for 30 min at room temperature. The samples were then infused into the nsDNA-carpeted flow cell at 5 µl/min for 15 min. At this point, the incoming components interacting with the nsDNA-carpet would be approaching a steady state. The carpet-bound protein complexes were then washed with a buffer containing ATP without ParA$_{pSM}$, ParB$_{pSM}$, or parS$_{pSM}$. ParA$_{pSM}$-GFP-ATP in the absence of ParB$_{pSM}$ reached saturation on the nsDNA-carpet in 5–10 min and then was released from the carpet slowly during the wash phase with dissociation kinetics that fit a double exponential function, majority fraction dissociating slowly ($k_{off\text{-}fast}$=1.7 ± 1.1 min$^{-1}$ [~7%], $k_{off\text{-}slow}$=0.014 ± 0.002 min$^{-1}$, N = 2) (**Figure 2—figure supplement 2B**, top) as observed starting with lower ParA$_{pSM}$-GFP density (**Figure 2B**).

When ParA$_{pSM}$-GFP was mixed with ParB$_{pSM}$ and ATP in the absence of parS$_{pSM}$ and infused into the flow cell, ParA$_{pSM}$-GFP-ATP reached near saturation density on the nsDNA-carpet in ~10 min. During the wash phase, ParA$_{pSM}$-GFP dissociated from the nsDNA-carpet with double exponential kinetics ($k_{off\text{-}fast}$ = 2.3 ± 0.5 min$^{-1}$, $k_{off\text{-}slow}$ = 0.057 ± 0.01 min$^{-1}$, N = 4) as in the absence of ParB$_{pSM}$ except for the several fold faster slow phase $k_{off}$ and increased fraction of faster-dissociating population (~30%) (**Figure 4Ad**). When the reaction mixture also contained 3R-parS$_{pSM}$, ParA$_{pSM}$-GFP dissociation kinetics were similar to the above, except reduced fraction of the faster dissociating ParA$_{pSM}$-GFP population (**Figure 4Ac**). In these experiments, ParB$_{pSM}$ first accumulated on the nsDNA-carpet to roughly two-thirds of the density of ParA$_{pSM}$. When washing started, the majority (~75%) of ParB$_{pSM}$ dissociated quickly (~4 or ~7 min$^{-1}$, with or without 3R-parS$_{pSM}$). Thus, even in the presence of 3R-parS$_{pSM}$ fragment, most ParB$_{pSM}$ dissociates from ParA$_{pSM}$ bound to nsDNA-carpet within half of a minute (**Figure 4Ac and B**).

When ParA$_{pSM}$-GFP and ParB$_{pSM}$ were preincubated with 7R-parS$_{pSM}$, the steady-state ParA$_{pSM}$ accumulation on the nsDNA-carpet was suppressed by nearly 50%, and ParB$_{pSM}$ and ParA$_{pSM}$ accumulated on the nsDNA-carpet at ~1A:1.2B ratio. When the wash was started, the two proteins dissociated from the carpet maintaining roughly constant protein stoichiometry with kinetics that can be fit to a double-exponential curve ($k_{off\text{-}fast\text{-}A}$ [~73%] 0.92 ± 0.18 min$^{-1}$, $k_{off\text{-}slow\text{-}A}$ 0.075 ± 0.04 min$^{-1}$, and $k_{off\text{-}fast\text{-}B}$ [~70%] 1.1 ± 0.14 min$^{-1}$, $k_{off\text{-}slow\text{-}B}$ 0.13 ± 0.03 min$^{-1}$, N = 4; **Figure 4Aa**). Thus, before ATP hydrolysis

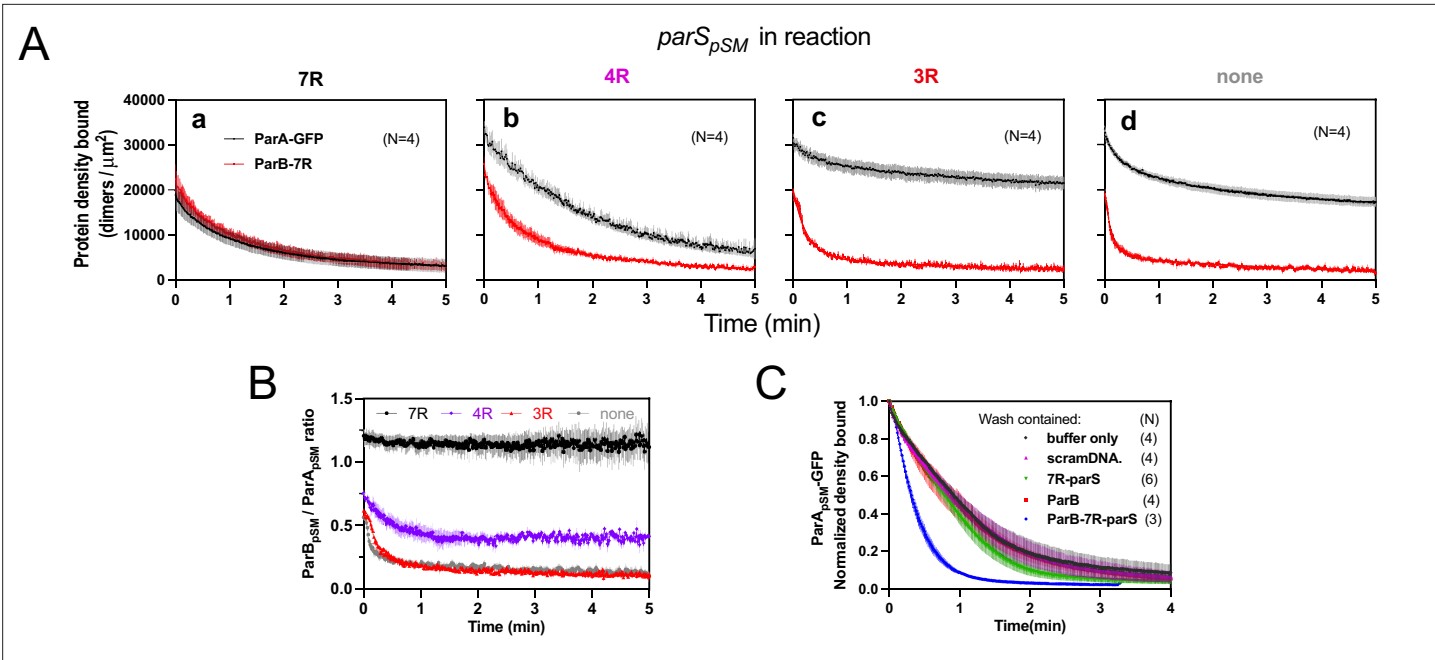

**Figure 4.** Stable ParA$_{pSM}$-ParB$_{pSM}$ complex is formed prior to ATP hydrolysis in the presence of functionally active *parS$_{pSM}$*. (**A**) ParA$_{pSM}$-GFP (1 µM) and ParB$_{pSM}$-Alexa647 (1% labeled, 1 µM) were preincubated with ATP (1 mM) plus (**a**) 7R-, (**b**) 4R-, (**c**) 3R-*parS$_{pSM}$*, or (**d**) without *parS$_{pSM}$*. The preincubated sample was infused into the nsDNA-carpeted flow cell at 5 µl/min for 15 min and then the solution flowing over the observation area was switched to a buffer containing ATP (t = 0). Fluorescence signals of ParA$_{pSM}$ (black) and ParB$_{pSM}$ (red) were converted to protein density on the nsDNA-carpet (dimers per µm²) and plotted. Each time point represents mean and SEM (vertical spread) of N experiments. (**B**) Time course of the carpet-bound ParB$_{pSM}$:ParA$_{pSM}$ molar ratio for the four panels of (**A**) (7R: black; 4R: purple; 3R: red; none: gray). (**C**) Continued supply of ParB$_{pSM}$ and *parS$_{pSM}$* in the washing solution is required to sustain high rate of ParA$_{pSM}$ disassembly. The experiment in the presence of 7R-*parS$_{pSM}$* in panel (**Aa**) was repeated with addition of ParB$_{pSM}$ and/or 7R-*parS$_{pSM}$* in the wash solution. Fluorescence signals of ParA$_{pSM}$ were normalized to the value at t = 0. Each time point represents mean and standard errors of mean (SEM) (vertical spread) of N experiments.

The online version of this article includes the following source data for figure 4:

**Source data 1.**

disassembles ParA$_{pSM}$ dimer from the nsDNA-carpet, ParA$_{pSM}$-ParB$_{pSM}$ protein interaction appears to be significantly more stable in the presence of 7R-*parS$_{pSM}$* compared to the complex involving 3R-*parS$_{pSM}$* (*Figure 4B*).

In the presence of 4R-*parS$_{pSM}$*, carpet binding and dissociation dynamics of the two proteins were qualitatively similar to those in the presence of 7R-*parS$_{pSM}$*, except for the following differences: 4R-*parS$_{pSM}$* did not suppress ParA$_{pSM}$ binding to the nsDNA-carpet as effectively as 7R-*parS$_{pSM}$*, the ParB$_{pSM}$/ParA$_{pSM}$ ratio at the end of 15 min sample infusion was ~0.75, which dropped to ~0.5 within 1 min of washing after which the ratio remained constant. Observed $k_{off}$ were ParA$_{pSM}$ ($k_{off\text{-}fast\text{-}A}$ [60%] 0.67 ± 0.14 min$^{-1}$, $k_{off\text{-}slow\text{-}A}$ 0.17 ± 0.05 min$^{-1}$; and $k_{off\text{-}fast\text{-}B}$ [66%] 1.9 ± 0.2 min$^{-1}$, $k_{off\text{-}slow\text{-}B}$ 0.23 ± 0.02 min$^{-1}$, N = 4; *Figure 4Ab*). It appears that ParA$_{pSM}$ bound to the nsDNA-carpet interacts slightly less stably with ParB$_{pSM}$ in the complex assembled with 4R-*parS$_{pSM}$* compared to those assembled with 7R-*parS$_{pSM}$*, but it was still significantly more stable compared to the majority of the ParA$_{pSM}$-ParB$_{pSM}$ complexes that accumulate in the presence of 3R-*parS$_{pSM}$* or without *parS$_{pSM}$* DNA.

In this experiment, the absence of ParB$_{pSM}$ and 7R- or 4R-*parS$_{pSM}$* in the wash solution flowing over the nsDNA-carpet appeared to have caused a slowdown of dissociation of the ParA$_{pSM}$-ParB$_{pSM}$ complexes from nsDNA compared to the experiments of *Figures 2B and 3A*. This prompted us to examine whether partial loss of the *parS$_{pSM}$* DNA and/or ParB$_{pSM}$ from the complex during the washing caused this kinetic change. We repeated the experiments of *Figure 4Aa* with the addition of ParB$_{pSM}$ (1 µM) and/or 7R-*parS$_{pSM}$* DNA fragment (500 nM consensus repeats) in the washing solution. The addition of ParB$_{pSM}$ and 7R-*parS$_{pSM}$* in the washing solution significantly accelerated ParA$_{pSM}$ $k_{off}$ to 2.5 ± 0.25 min$^{-1}$ (N = 3), a slightly higher level than seen in *Figure 3Aa* (1.8 ± 0.05 min$^{-1}$), which might have been an underestimate due to the preceding slow complex assembly steps. The addition of ParB$_{pSM}$ or

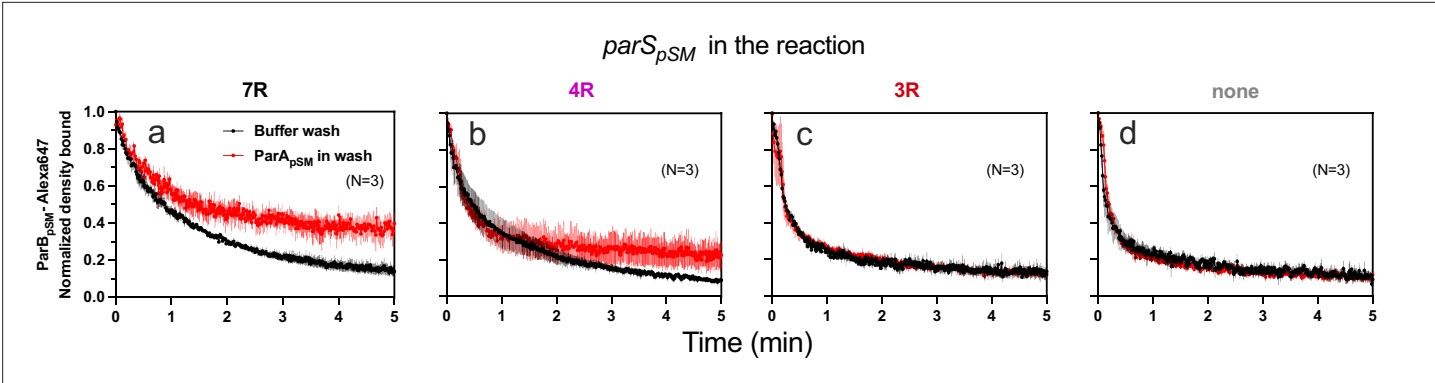

**Figure 5.** ParA$_{pSM}$-ATP stabilizes functionally active *parS$_{pSM}$*-bound ParB$_{pSM}$ on the nsDNA-carpet. Complexes containing ParA$_{pSM}$-GFP, and ParB$_{pSM}$-Alexa647 bound to the DNA-carpet together with different *parS$_{pSM}$* DNA fragments, (**a**) 7R, (**b**) 4R, (**c**) 3R, or (**d**) without *parS$_{pSM}$*, were washed with either a buffer containing 1 mM ATP (black) or the same buffer containing ParA$_{pSM}$-GFP (0.5 μM, red), in addition to ATP. The ParB$_{pSM}$-Alexa647 dissociation curves normalized to the protein density at t = 0 were plotted. Each time point represents mean and standard errors of mean (SEM) (vertical spread) of N experiments.

The online version of this article includes the following source data for figure 5:

**Source data 1.**

---

7R-*parS$_{pSM}$* separately to the washing solution did not have significant effects on the ParA$_{pSM}$ dissociation kinetics (**Figure 4C**). Thus, partial loss of ParB$_{pSM}$-*parS$_{pSM}$* complex from the nsDNA-bound ParA$_{pSM}$ at the onset of the wash (7R- and 4R-*parS$_{pSM}$*) and also during the wash (4R-*parS$_{pSM}$*) appears to have compromised the efficiency of ParA$_{pSM}$ dissociation from nsDNA in the experiments of **Figure 4Aa,b**. This indicates that not all the ParB$_{pSM}$-*parS$_{pSM}$* complexes that can contribute to dissociation of ParA$_{pSM}$ from nsDNA are stably retained via ParA$_{pSM}$ to the nsDNA-carpet through the course of reaction.

## ParA$_{pSM}$-ATP in the wash solution stabilizes functional ParB$_{pSM}$-*parS$_{pSM}$* complexes associated with carpet-bound ParA$_{pSM}$

When preformed ParA$_{pSM}$-ParB$_{pSM}$ complexes bound to the nsDNA-carpet in the presence of 7R- or 4R-*parS$_{pSM}$* DNA were washed with buffer containing ATP-activated ParA$_{pSM}$-GFP (0.5 μM), freshly arriving ParA$_{pSM}$-ATP dimers on the nsDNA-carpet replacing the dissociating ParA$_{pSM}$ partially stabilized ParB$_{pSM}$ (**Figure 5a, b**). This observation indicates that when ParA$_{pSM}$ dissociates from nsDNA following ATP hydrolysis, some fraction of ParB$_{pSM}$ in the presence of 7R- or 4R-*parS$_{pSM}$* that induced ATP hydrolysis can capture the newly arriving ParA$_{pSM}$-ATP dimers on the nsDNA-carpet. The complex containing 7R-*parS$_{pSM}$* was more efficient for recapture than the 4R-*parS$_{pSM}$* complex. We propose larger numbers of ParB$_{pSM}$ dimers complexed with *parS$_{pSM}$* with more sequence repeats can interact simultaneously with multiple ParA$_{pSM}$-nsDNA mini-filaments, more readily capturing newly arrived ParA$_{pSM}$-ATP dimers at nearby nsDNA sites. Evidence supporting such multivalent interactions between nsDNA-bound ParA$_{pSM}$ and *parS$_{pSM}$*-bound ParB$_{pSM}$ has been reported previously (**Pratto et al., 2009**). ParB$_{pSM}$ stabilization by replenishment of ParA$_{pSM}$-ATP was not observed in the presence of 3R-*parS$_{pSM}$* or in the absence of *parS$_{pSM}$*, consistent with the notion that ParA$_{pSM}$-ParB$_{pSM}$ interactions under these conditions are unstable (**Figure 5c,d**).

## Discussion

Using a combination of steady-state ATPase assays and nsDNA dissociation kinetic measurements of ParA$_{pSM}$-ParB$_{pSM}$ complexes in the presence of a variety of modified *parS$_{pSM}$*-DNA fragments, we studied the molecular requirements for activation of ParA$_{pSM}$. This work revealed distinct *parS$_{pSM}$* structural requirements for efficient activation of ParA$_{pSM}$ ATPase by ParB$_{pSM}$ and a multistep assembly process for the active *parS$_{pSM}$*-ParB$_{pSM}$-ParA$_{pSM}$ complex. The *parS$_{pSM}$* DNA must contain at minimum four contiguous consensus sequence repeats to enable ParB$_{pSM}$ dimer binding in a state that induces ATP hydrolysis and accelerated dissociation of ParA$_{pSM}$ from nsDNA. The structural requirements for *parS$_{pSM}$* function assure high specificity of the centromere site for PC function. Considering the clear

biochemical functionality difference depending on the $parS_{pSM}$ sequence repeat number and the repeat contiguity, we believe in vivo plasmid stability would also require $parS_{pSM}$ site(s) at minimum four or possibly larger number of repeats in contiguous arrangements. However, this point remains to be experimentally confirmed. Considering that there are six copies of $parS_{pSM}$ sites in pSM19035, it also would be useful to learn the number of $parS_{pSM}$ sites required for the plasmid stability and whether this requirement depends on the plasmid copy number.

## ParB$_{pSM}$-*parS$_{pSM}$* recognition by ParA$_{pSM}$

The $parS_{pSM}$ sequence features required for the function of ParB$_{pSM}$, an RHH-ParB protein contrast with those for the HTH-ParB proteins. Whereas natural $parS_F$ sequence for the F-plasmid is composed of 12 repeats of the ParB$_F$ dimer binding sequence, a single copy of this consensus sequence is able to support faithful plasmid partition (*Biek and Shi, 1994*). Further, HTH-ParB proteins can spread around *parS* sites, forming large PCs containing many ParB molecules, most of which have moved away from *parS* sequence. In contrast, RHH-ParB proteins lack the CTPase domain that is critical for HTH-ParB spreading (*Soh et al., 2019*; *Jalal et al., 2020*; *Osorio-Valeriano et al., 2019*) and they cannot spread from *parS* sites to flanking DNA (*Pratto et al., 2009*). Systems involving RHH-ParB proteins perhaps evolved a fundamentally different system architecture for PC dynamics to accomplish robust partitioning of replicated plasmid copies. It is currently unclear what constitutes the defining feature of the ParB$_{pSM}$ molecules bound to four or more contiguous $parS_{pSM}$ heptad-sequence-repeats: we offer a possible scenario below. It is interesting to note that $parS_{pSM}$-dependence of ParA$_{pSM}$-ATPase activation is high compared to F-plasmid Par system involving HTH-ParB, which exhibits either only less than twofold higher ATPase activation in the presence of $parS_F$ (in the absence of CTP) or no difference of the maximum ATP turnover rate in the presence of CTP (*Ah-Seng et al., 2009*; *Taylor et al., 2021*). This is reasonable for HTH-ParB systems; most ParB molecules in the PC are not *parS*-associated at the time it interacts and activates the ParA$_F$-ATPase and only limited number of ParB$_F$ molecules are outside of PCs. In contrast, fewer PC-associated RHH-ParB molecules and perhaps comparatively more non-PC-associated RHH-ParB molecules exist inside a cell, necessitating tighter *parS*-dependent ATPase activation for the nonspreading ParB systems.

## Assembly/disassembly dynamics of ParAB$_{pSM}$ complex on the nsDNA

ParA$_{pSM}$-ATP is slow to dissociate from the nsDNA-carpet in the presence of ParB$_{pSM}$ alone, or ParB$_{pSM}$-3R-*parS$_{pSM}$* complex in the wash solution. We showed that ParB$_{pSM}$ can quickly interact with nsDNA-bound ParA$_{pSM}$ in the absence of functional $parS_{pSM}$ DNA (*Figure 3Ac*). However, this ParA$_{pSM}$-ParB$_{pSM}$ interaction does not fully activate the ATPase (*Figure 1*). All wash curves of ParA$_{pSM}$-ATP dimers bound to nsDNA-carpet under conditions that do not trigger efficient ATPase activation exhibited double-exponential dissociation kinetics, although the small amplitude of the fast dissociation phase, typically ~5%, made quantitative comparison difficult (*Figures 2B and 3A*). The fast dissociation phase had a time scale of less than 1 min, while the majority fraction dissociated much slower under conditions that do not activate the ATPase. This indicated that the carpet-bound ParA$_{pSM}$ dimers were composed of two distinct state populations, transitions between which are slow. ParB$_{pSM}$ with or without 3R-*parS$_{pSM}$*, after the initial rapid binding to nsDNA-bound ParA$_{pSM}$, also dissociated with double-exponential kinetics; a fraction of the carpet-associated ParB$_{pSM}$ quickly dissociated within 1 min before settling to a quasi-steady state with the constant supply of ParB$_{pSM}$ or ParB$_{pSM}$-3R-*parS$_{pSM}$* complex in the wash solution and slowly dissociating ParA$_{pSM}$ on the nsDNA-carpet (*Figure 3Ac,d*). We speculate that this small initial binding overshoot reflects ParB$_{pSM}$ association to the less populated fast-dissociating fraction of the carpet-bound ParA$_{pSM}$ dimers, which presumably are in a state with faster ParB$_{pSM}$-association rate, lower nsDNA affinity, and perhaps closer to the ATP hydrolysis-competent state.

When ParA$_{pSM}$-ATP-ParB$_{pSM}$ complexes bound to the nsDNA-carpet in a steady state with or without 3R-*parS$_{pSM}$* was washed with a simple buffer (*Figure 4Ac,d*), a double-exponential dissociation of ParA$_{pSM}$ was again observed with increased fraction (up to ~30%) of the fast-dissociating population. This supports the notion that the fast-dissociating ParA$_{pSM}$ population is in a state closer to, but not committed to, ATP hydrolysis. We hypothesize that this ParA$_{pSM}$ transitory state becomes more populated during preincubation with ParB$_{pSM}$ on the nsDNA-carpet accounting for the moderate stimulation of the DNA-bound ParA$_{pSM}$ ATPase by ParB$_{pSM}$ or ParB$_{pSM}$-3R-*parS$_{pSM}$* complex at steady state (*Figure 1A*). However, this 'fast' dissociation does not depend on ATP hydrolysis, considering the

ATPase-defective mutant ParA$_{pSM}$$^{D60E}$ also exhibits clear double-exponential dissociation (*Figure 2—figure supplement 2B*, top, *Figure 2—figure supplement 3B*).

When ParA$_{pSM}$-ATP prebound to nsDNA-carpet in the flow cell was washed with a solution containing fully functional 7R-*parS$_{pSM}$*-ParB$_{pSM}$ complexes, accelerated release of ParA$_{pSM}$ from nsDNA was observed (*Figure 2B*, *Figure 3Aa,e,i*). However, the initial ParA$_{pSM}$ dissociation kinetics resembled the ParB$_{pSM}$ wash without fully active *parS$_{pSM}$* and accelerated disassembly started with a delay. This clear transition of the dissociation mode indicates the initial complex between the nsDNA-bound ParA$_{pSM}$ and *parS$_{pSM}$*-bound ParB$_{pSM}$ does not immediately trigger ATP hydrolysis. Rather a series of steps must take place subsequent to the initial association of the two protein-DNA complexes to trigger ATP hydrolysis and complex disassembly. This process appears to involve participation of additional ParB$_{pSM}$ dimers and/or *parS$_{pSM}$* beyond the initial complex, considering that the delay time before the accelerated ParA$_{pSM}$ dissociation phase depends on their concentration in the wash solution while clear biphasic kinetic feature was observed even at limiting ParB$_{pSM}$ concentrations.

The ParA$_{pSM}$-ParB$_{pSM}$ interaction is not stable in the absence of 7R- or 4R- *parS$_{pSM}$* (*Figure 4B*). In contrast, interaction between nsDNA-bound ParA$_{pSM}$ and ParB$_{pSM}$ in the presence of 7R-*parS$_{pSM}$* is significantly more stable prior to ATP hydrolysis-dependent dissociation of ParA$_{pSM}$ from nsDNA (*Figure 4Aa*). Thus, ParA$_{pSM}$-ATP interacts with ParB$_{pSM}$ in the presence of fully active *parS$_{pSM}$* in a distinct manner than in its absence. For simplicity, we propose this transition to stably interacting ParA$_{pSM}$-ParB$_{pSM}$ complex bound to nsDNA is the limiting step necessary before ATP hydrolysis-dependent acceleration of ParA$_{pSM}$ dissociation. Slower $k_{off}$ of ParA$_{pSM}$ from nsDNA in the experiments of *Figure 4Aa,b* compared to those in *Figure 2B* and *Figure 3Aa,b*, and recovery of faster $k_{off}$ by the replenishment of ParB$_{pSM}$ and *parS$_{pSM}$* in the wash solution (*Figure 4C*) indicated that ParA$_{pSM}$-ParB$_{pSM}$ complex must turn over to maintain high ParA$_{pSM}$ $k_{off}$. This suggests that not all the nsDNA-bound ParA$_{pSM}$ dimers are interacting with ParB$_{pSM}$ in the state committed for ATPase activation in the initial set of active complexes that form on the nsDNA-carpet.

## A model for ParB$_{pSM}$ activation of ParA$_{pSM}$-ATPase

Without further experimental constraints, our consideration of the mechanism of ParA$_{pSM}$-ATPase activation by ParB$_{pSM}$ remains speculative. Unlike other members of ParA-family of ATPases studied before, such as ParA$_F$ (*Taylor et al., 2021*) or MinD (*Vecchiarelli et al., 2016*), simple interaction of ParB$_{pSM}$-N-terminal domain, ParB$_{pSM}$$^{1-27}$, with ParA$_{pSM}$ dimers did not efficiently activate the ATPase (*Figure 1—figure supplement 1B*). Because of the helical nature of the ParB$_{pSM}$-*parS$_{pSM}$* complex (*Weihofen et al., 2006*) and ParA$_{pSM}$-nsDNA mini-filament formation (*Pratto et al., 2009*), two ParA$_{pSM}$ dimers within a mini-filament might be approximately in position to interact with ParB$_{pSM}$ dimers separated by two intervening *parS$_{pSM}$* sequence repeats on one face of the helical filament. Let us assume, however, that the pitch and/or the angular arrangements of these two pairs of protein dimers are not in perfect match allowing only partial interaction between ParA$_{pSM}$ dimers and ParB$_{pSM}$ dimers in the basal state structures of the two mini-filaments (*Figure 6A*). Then, the establishment of divalent interactions and full engagement between the two protein-DNA complexes perhaps would force distortions of the structures of both of the protein-DNA mini-filaments (*Figure 6B*). The force imposed upon ParA$_{pSM}$-mini-filament might act as a steppingstone for the conformational change of ParA$_{pSM}$ necessary for ATPase activation. Such a scenario explains the requirement for the fourth *parS$_{pSM}$* consensus sequence repeat for the ATPase activation. We propose the specific mechanical properties of the contiguous ParB$_{pSM}$-*parS$_{pSM}$* mini-filament, likely different from gapped mini-filaments, is required for efficient ATPase activation, explaining the requirement for the *parS$_{pSM}$* sequence repeat contiguity. The divalent interactions between one ParB$_{pSM}$-*parS$_{pSM}$* mini-filament with a ParA$_{pSM}$-nsDNA mini-filament, with a non-interacting middle segment separating the two interacting pairs, might allow the ParB$_{pSM}$-*parS$_{pSM}$* complex to semi-processively activate multiple ParA$_{pSM}$-nsDNA complexes in succession by an inchworm-like transfer to a new ParA$_{pSM}$-nsDNA mini-filament. The ability of the otherwise dissociating 7R-*parS$_{pSM}$*-ParB$_{pSM}$ complex after ATPase activation and dissociation of one partner ParA$_{pSM}$ mini-filament to recapture a freshly arriving ParA$_{pSM}$-ATP dimers on the nsDNA-carpet (*Figure 5a*) is consistent with such a possibility.

The concentration mismatch between ParB$_{pSM}$ and *parS$_{pSM}$* needed for full activation of ParA$_{pSM}$ ATPase (*Figure 1D*) suggests additional details; the functional *parS$_{pSM}$* DNA might be needed only to deliver ParB$_{pSM}$ dimers onto ParA$_{pSM}$-nsDNA mini-filament generating a complex committed to ATP

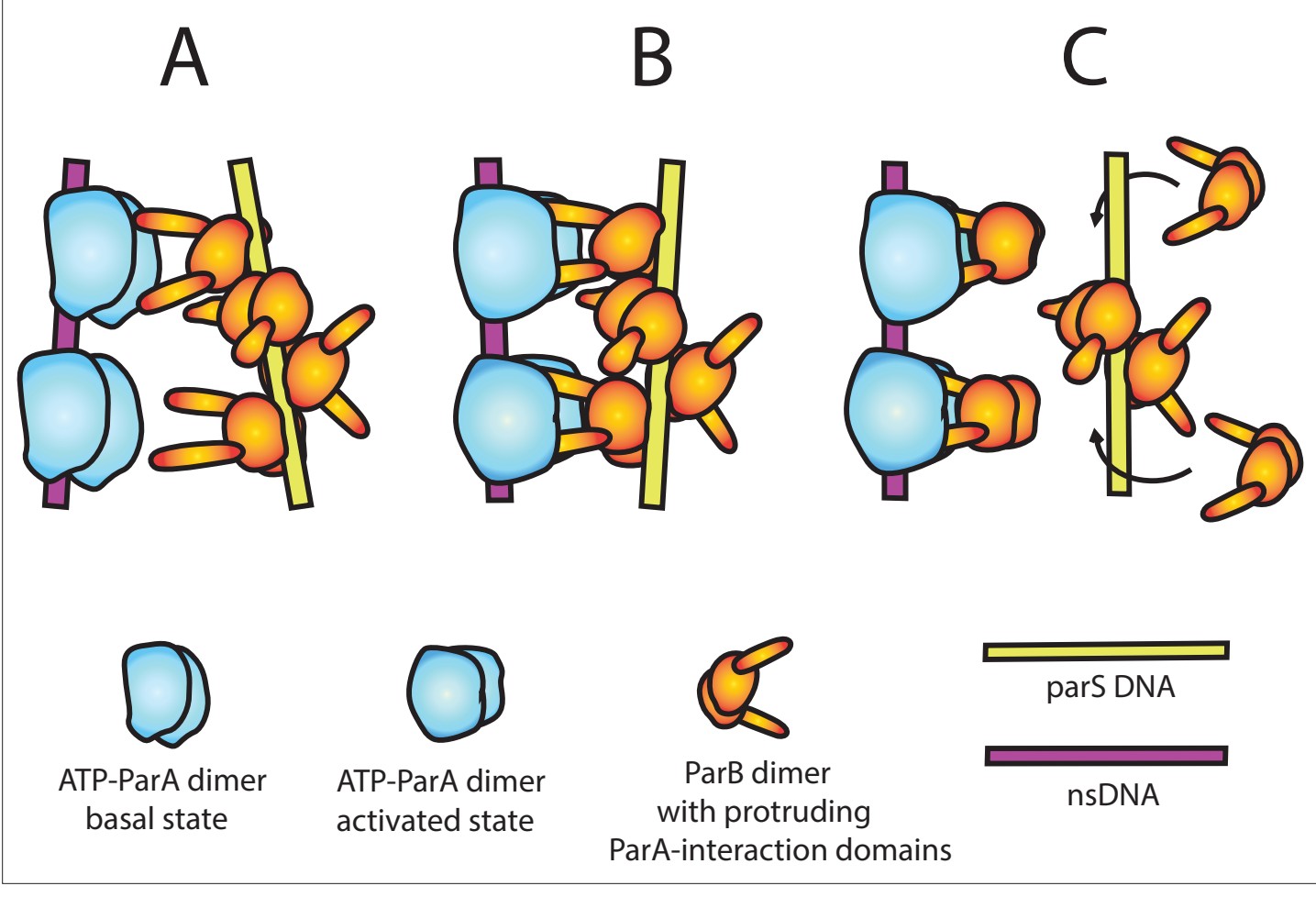

**Figure 6.** A model of nsDNA-bound ParA$_{pSM}$-ATPase activation by *parS$_{pSM}$*-bound ParB$_{pSM}$. (**A**) ParA$_{pSM}$-ATP dimers bound to nsDNA in their basal state can interact with ParA-activation domains protruding from ParB$_{pSM}$ dimers bound to *parS$_{pSM}$* sequence repeats. However, multiple ParA$_{pSM}$ dimers in a mini-filament cannot simultaneously interact with ParB$_{pSM}$ dimers bound to a set of repeated sequence copies within a *parS$_{pSM}$* site and the two proteins dissociate quickly. We propose the interacting pair of proteins at this stage are not fully engaged and these ParA$_{pSM}$ dimers are not in the ATPase-activated state. Here, we assume two ParA-interacting domains belonging to one ParB$_{pSM}$ dimer binds one ParA$_{pSM}$ dimer. (**B**) We propose torsional (or other conformational) thermal Brownian dynamics of the mini-filaments allow the ParB$_{pSM}$ dimer at the fourth position to establishes interaction with another ParA$_{pSM}$ dimer, locking in the non-equilibrium conformation of the individual mini-filaments prior to the formation of this second bridge. The distortion promotes conformational transition of the ParA$_{pSM}$ dimers to the ATPase active state with fully engaged ParB$_{pSM}$. (**C**) The conformational transition also destabilizes ParB$_{pSM}$-*parS* interaction, releasing the *parS$_{pSM}$* DNA from the activated nsDNA-ParA$_{pSM}$-ParB$_{pSM}$ complex prior to ATP hydrolysis and disassembly of the complex. This allows reloading of fresh ParB$_{pSM}$ to the released *parS$_{pSM}$*, which recycles to disassemble the remaining ParA$_{pSM}$ on the nsDNA. Meanwhile, fully engaged ParB$_{pSM}$ dimers left on the ParA$_{pSM}$ dimers cause ATP hydrolysis and disassemble ParA$_{pSM}$ dimers from nsDNA.

hydrolysis. After ParB$_{pSM}$ delivery, *parS$_{pSM}$* might release the delivered ParB$_{pSM}$ dimers, perhaps helped by the mini-filament distortion discussed above, without waiting for ATP hydrolysis (**Figure 6C**). The emptied consensus sequence on the *parS$_{pSM}$* would then be quickly recharged with free ParB$_{pSM}$ dimers in solution to activate another ParA$_{pSM}$-nsDNA complex, explaining the higher concentration demand for ParB$_{pSM}$ over *parS$_{pSM}$* for ATPase activation. The requirement for *parS$_{pSM}$* and ParB$_{pSM}$ replenishment from the washing solution to sustain maximum $k_{off}$ of the nsDNA-bound ParA$_{pSM}$ dimers (**Figure 4C**) supports the notion that the initial active ParA$_{pSM}$-ParB$_{pSM}$-*parS$_{pSM}$* complex assembled on the nsDNA perhaps does not induce ATP hydrolysis of all the ParA$_{pSM}$ dimers within the complex. Full disassembly of the complex likely involves release of the *parS$_{pSM}$* fragment, which gets reloaded with ParB$_{pSM}$ in solution and returns to the ParA$_{pSM}$ dimers left on the nsDNA. This local recycling of ParB$_{pSM}$ and *parS$_{pSM}$* would become inefficient when ParB$_{pSM}$ and *parS$_{pSM}$* are removed from reaction by the buffer wash. We note that dependence of the accelerated phase of ParA$_{pSM}$ $k_{off}$ on the concentration of

the $parS_{pSM}$-ParB$_{pSM}$ complex (**Table 1**) is consistent with the above observation. In vivo, local off-rate of ParA$_{pSM}$ from nucleoid where a PC is located is likely to be significantly faster than observed here, considering the local density of the nucleoid-bound ParA$_{pSM}$ near a PC would be lower and the ParB$_{pSM}$-$parS_{pSM}$ concentration of a PC, with six copies of $parS$ sites, much higher compared to the conditions in this study.

Earlier we described our model considering one ParB$_{pSM}$-$parS_{pSM}$ mini-filament acting on one ParA$_{pSM}$ cluster on nsDNA for simplicity. However, this does not immediately explain the clearly biphasic disassembly kinetics of the nsDNA-bound ParA$_{pSM}$ during washing with ParB$_{pSM}$-$parS_{pSM}$ complex (**Figure 2B**, **Figure 3Aa,e,i**). This kinetics is highly reminiscent of activation and membrane dissociation kinetics of membrane-bound MinD-ATPase, a member of ParA-ATPase family, when washed by MinE-containing buffer (**Vecchiarelli et al., 2016**), which has been proposed to reflect requirement for two MinE dimers for the activation of one MinD dimer. Thus, we suspect two ParB$_{pSM}$-$parS_{pSM}$ mini-filaments may need to cooperate from two sides of one ParA$_{pSM}$ cluster on nsDNA, and the second binding is kinetically limiting.

Lastly, irrespective of the details of the model, we need to pay attention to a few aspects of the pSM19035 Par system in the context of the diffusion-ratchet concept. As mentioned in the 'Introduction,' plasmid or chromosome partition by a diffusion-ratchet mechanism requires the balance between the stability of each ParA/ParB-mediated cargo-nucleoid bridge and the number of these bridges per cargo. In the pSM19035 system, prior to ATPase-activating nsDNA-ParA$_{pSM}$-ParB$_{pSM}$-$parS_{pSM}$ complex assembly, the ParA$_{pSM}$-ParB$_{pSM}$ link in the bridge is expected to have a stability close to that in the presence of inactive 3R-$parS_{pSM}$ ($k_{off}$ = ~4 min$^{-1}$, **Figure 4Ac**). This stability is comparable to that of F-plasmid ParA$_F$- ParB$_F$ bridge between the nucleoid and cargo, which we believe to involve one ParA$_F$ dimer, and therefore DNA-carpet-bound ParA$_F$ FRAP rate in the presence of ParB$_F$ (~4 min$^{-1}$, **Vecchiarelli et al., 2013**) would approximate the stability of the bridge. In contrast, we believe that the effective stability of a unit of the cargo-nucleoid bridge by the ATPase-activating nsDNA-ParA$_{pSM}$-ParB$_{pSM}$-$parS_{pSM}$ complex containing many ParA$_{pSM}$ dimers and ParB$_{pSM}$ dimers is likely to be significantly higher. If one assumes processive rounds of ATPase activation for subsets of ParA$_{pSM}$ dimers within a complex as considered in our model, individual cargo-nucleoid bridge would have substantially longer lifetime than individual ParA$_{pSM}$ dimer dissociation rate constant of up to ~2 min$^{-1}$ observed by the experiments of **Figure 3**. At the same time, the number of individual units of bridges to the nucleoid per cargo would be very small compared to up to a few hundred in cases of HTH-ParB plasmid systems. Long-range transfer of the PC from one ParA$_{pSM}$ cluster to another via simultaneous multiple ParA$_{pSM}$ cluster interactions involving nucleoid DNA looping without bridge dissolution discussed above would further extend the lifetime of individual bridges, enhancing the effective PC diffusion suppression capability of very small number of bridges without blocking the PC motion. In addition, very slow dissociation of ParA$_{pSM}$ dimers from nsDNA prior to formation of fully active complex with a PC ($k_{off}$ < 0.1 min$^{-1}$, **Figures 2 and 3**) compared to ParA$_F$ or ParA$_{P1}$ ($k_{off}$ = ~5 min$^{-1}$, **Hwang et al., 2013**, **Vecchiarelli et al., 2013**) suggests their very slow diffusion on the nucleoid (practically no hopping). Together with expected slow overall ATP hydrolysis rate, it predicts slow ParA$_{pSM}$ depletion zone development and refilling. While these parameter shifts compared to the F- or P1-Par systems are generally speaking in the mutually compensatory directions (**Hu et al., 2015**; **Hu et al., 2017**), whether they are balanced or not needs to be carefully evaluated as more details of the mechanism come to light in future.

This study revealed a puzzlingly unique $parS_{pSM}$ DNA site structure required for the assembly of the fully active ParB$_{pSM}$-$parS_{pSM}$ PC in the pSM19035 ParABS system. We propose one possible scenario to explain our findings. However, the model presented here is perhaps not the only possible explanation. Further studies are needed to support or refute the model to advance our mechanistic understanding of this family of partition systems. Direct examination of the effects of the PC torsional strain would be needed to test our current model. Further refinement of the cell-free partition reaction system is hoped to greatly assist our mechanistic understanding of the rich variations of the prokaryotic chromosome/plasmid partition systems.

# Materials and methods

## Key resources table

| Reagent type (species) or resource | Designation | Source or reference | Identifiers | Additional information |
|---|---|---|---|---|
| Strain, strain background (*Escherichia coli*) | BL21 DE3 AI | Invitrogen | C607003 | Protein expression strain |
| Recombinant DNA reagent | pET11a | EMD Millipore | 9436 | Protein expression vector |
| Recombinant DNA reagent | pT712ω | *Welfle et al., 2005* | | ParB$_{pSM}$ overexpression plasmid |
| Recombinant DNA reagent | pCB746 (pT712 vector) | *Pratto et al., 2008* | | ParA$_{pSM}$ overexpression plasmid |
| Recombinant DNA reagent | pCB755 (pT712 vector) | *Pratto et al., 2008* | | ParA$_{pSM}$$^{D60A}$ overexpression plasmid |
| Recombinant DNA reagent | pET11a-ParA$_{pSM}$$^{D60E}$ | This work | | ParA$_{pSM}$$^{D60E}$ overexpression plasmid |
| Recombinant DNA reagent | pCB1033 (pT712 vector) | This work | | ParB$_{pSM}$-GCE overexpression plasmid |
| Recombinant DNA reagent | pET11a-ParA$_{pSM}$-EGFP | This work | | ParA$_{pSM}$-EGFP overexpression plasmid |
| Recombinant DNA reagent | pET11a-ParA$_{pSM}$$^{D60A}$-EGFP | This work | | ParA$_{pSM}$$^{D60A}$-EGFP overexpression plasmid |
| Recombinant DNA reagent | pET11a-ParA$_{pSM}$$^{D60E}$-EGFP | This work | | ParA$_{pSM}$$^{D60E}$-EGFP overexpression plasmid |
| Sequence-based reagent | Scrambled 55mer DNA oligo + strand | This work | | GGGATCAAACACTTGATAGACAAGTCTTTGACCTAATTGTGAAAATTATGAAGGG |
| Sequence-based reagent | Scrambled 55mer DNA oligo - strand | This work | | CCCTTCATAATTTTCACAATTAGGTCAAAGACTTGTCTATCAAGTGTTTGATCCC |
| Sequence-based reagent | 7R *parS* DNA oligo + strand | This work | | GGGAATCACAAATCACAAGTGATTAATCACAAATCACTTGTGATTTGTGATTGGG |
| Sequence-based reagent | 7R *parS* DNA oligo - strand | This work | | CCCAATCACAAATCACAAGTGATTTGTGATTAATCACTTGTGATTTGTGATTCCC |
| Sequence-based reagent | 6R *parS* DNA oligo + strand | This work | | GGGAATCACAAGTGATTAATCACAAATCACTTGTGATTTGTGATTGGG |
| Sequence-based reagent | 6R *parS* DNA oligo - strand | This work | | CCCAATCACAAATCACAAGTGATTTGTGATTAATCACTTGTGATTCCC |
| Sequence-based reagent | 5R *parS* DNA oligo + strand | This work | | GGGAATCACAAATCACAAGTCACTTGTGATTTGTGATTGGG |

*Continued on next page*

*Continued*

| Reagent type (species) or resource | Designation | Source or reference | Identifiers | Additional information |
|---|---|---|---|---|
| Sequence-based reagent | 5R *parS* DNA oligo - strand | This work | | CCCAATCACAAATCACAAGTGATTTGTGATTTGTGATTCCC |
| Sequence-based reagent | 4R(1) *parS* DNA oligo + strand | This work | | GGGAATCACAAATCACTTGTGATTTGTGATTGGG |
| Sequence-based reagent | 4R(1) *parS* DNA oligo - strand | This work | | CCCAATCACAAATCACAAGTGATTTGTGATTCCC |
| Sequence-based reagent | 4R(2) *parS* DNA oligo + strand | This work | | GGGAATCACTTATCACAAGTGATTAATCACTGGG |
| Sequence-based reagent | 4R(2) *parS* DNA oligo - strand | This work | | CCCAGTGATTAATCACTTGTGATAAGTGATTCCC |
| Sequence-based reagent | 4R(3) *parS* DNA oligo + strand | This work | | GGGAATCACTTATCACAAATCACAAATCACTGGG |
| Sequence-based reagent | 4R(3) *parS* DNA oligo - strand | This work | | CCCAGTGATTTGTGATTTGTGATAAGTGATTCCC |
| Sequence-based reagent | 3R-2nc-3R *parS* DNA oligo + strand | This work | | GGGAATCACAAATCACAAATCACATCATAGTTCATAGTTGTGATTTGTGATTTGTGATTGGG |
| Sequence-based reagent | 3R-2nc-3R *parS* DNA oligo - strand | This work | | CCCAATCACAAATCACAAATCACAACTATGAACTATGATGTGATTTGTGATTTGTGATTCCC |
| Sequence-based reagent | 3R-1nc-3R *parS* DNA oligo + strand | This work | | GGGAATCACAAATCACAAATCACATCATAGTTGTGATTTGTGATTTGTGATTGGG |
| Sequence-based reagent | 3R-1nc-3R *parS* DNA oligo - strand | This work | | CCCAATCACAAATCACAAATCACAACTATGATGTGATTTGTGATTTGTGATTCCC |
| Sequence-based reagent | 3R-1nc *parS* DNA oligo + strand | This work | | GGGAATCACAAATCACTTGTGATTTCATAGTGGG |
| Sequence-based reagent | 3R-1nc *parS* DNA oligo - strand | This work | | CCCACTATGAAATCACAAGTGATTTGTGATTCCC |
| Sequence-based reagent | 3R(1) *parS* DNA oligo + strand | This work | | GGGAATCACAAATCACTTGTGATTGGG |
| Sequence-based reagent | 3R(1) *parS* DNA oligo - strand | This work | | CCCAATCACAAGTGATTTGTGATTCCC |
| Sequence-based reagent | 3R(2) *parS* DNA oligo + strand | This work | | GGGAATCACTTATCACAAATCACAGGG |

*Continued on next page*

*Continued*

| Reagent type (species) or resource | Designation | Source or reference | Identifiers | Additional information |
|---|---|---|---|---|
| Sequence-based reagent | 3R(2) *parS* DNA oligo - strand | This work | | CCCTGTGATTTGTGATAAGTGATTCCC |
| Sequence-based reagent | 2R *parS* DNA oligo + strand | This work | | GGGAATCACTTGTGATTGGG |
| Sequence-based reagent | 2R *parS* DNA oligo - strand | This work | | CCCAATCACAAGTGATTCCC |
| Sequence-based reagent | 1R *parS* DNA oligo + strand | This work | | GGGAATCACTGGG |
| Sequence-based reagent | 1R *parS* DNA oligo - strand | This work | | CCCAGTGATTCCC |
| Peptide, recombinant protein | ParB$_{pSM}$$^{1-27}$ | This work | | MIVGNLGAQKAKRNDTPISAKKDIMGD |
| Peptide, recombinant protein | ParB$_{pSM}$$^{1-27}$$_{K10A}$ | This work | | MIVGNLGAQAAKRNDTPISAKKDIMGD |
| Peptide, recombinant protein | ParB$_{P1}$$^{1-30}$ | This work | | MSKKNRPTIGRTLNPSILSGFDSSSASGDR |
| Chemical compound, drug | [γ−$^{32}$P]ATP | PerkinElmer | NEG002A250UC | |
| Chemical compound, drug | Biotin-17-dCTP | Invitrogen | 65601 | |
| Chemical compound, drug | Terminal transferase | New England Biolabs | M0315 | |
| Chemical compound, drug | Alexa Fluor 488 C5 maleimide | Thermo Fisher | A10254 | |
| Chemical compound, drug | Alexa Fluor 594 C5 maleimide | Thermo Fisher | A10256 | |
| Chemical compound, drug | Alexa Fluor 647 C2 maleimide | Thermo Fisher | A20347 | |
| Chemical compound, drug | Antifoam Y-40 emulsion | Sigma | A5758 | |
| Chemical compound, drug | EDTA-free Sigmafast protease inhibitor cocktail tablet | Sigma | S8830 | |

*Continued on next page*

*Continued*

| Reagent type (species) or resource | Designation | Source or reference | Identifiers | Additional information |
| --- | --- | --- | --- | --- |
| Chemical compound, drug | DOPC | Avanti Polar Lipids | 850375P | |
| Chemical compound, drug | DOPE-Biotin | Avanti Polar Lipids | 870273C | |
| Chemical compound, drug | Biotin-14-dCTP | Thermo Fisher | 19518018 | |
| Software, algorithm | Prism 9 | GraphPad | Prism 9 | Used for curve fitting, and fitting parameters and their error estimation |
| Software, algorithm | MetaMorph 7 | Molecular Devices | MetaMorph 7 | Used for TIRF microscope data acquisition |
| Software, algorithm | ImageJ/Fiji | National Institutes of Health | ImageJ | Used for TIRF microscope image analysis |
| Other (instrument) | Prism type TIRF microscope | In-house; *Vecchiarelli et al., 2013*, *Ivanov and Mizuuchi, 2010* | | Used for ParAF-ParBF complex assembly–disassembly experiments |

## Materials availability

Newly generated materials from this study are available by request to the corresponding author, Kiyoshi Mizuuchi (kiyoshimi@niddk.nih.gov) until the lab stocks become exhausted or the lab group operation becomes terminated.

## Proteins, peptides, and DNA

Non-fluorescent $ParA_{pSM}$-His$_6$ (wild-type and mutants) was purified as previously described (*Pratto et al., 2008*). $ParA_{pSM}$-GFP-His$_6$ and $ParA_{pSM}^{D60E}$-GFP-His$_6$ were purified as described for $ParA_F$-GFP-His$_6$ (*Vecchiarelli et al., 2010*). $ParB_{pSM}$ wild-type and $ParB_{pSM}$-cys, which had three residues (-GCE) added at the C-terminal, were purified essentially as previously described with the addition of reducing agent (2 mM DTT) in the buffers 50 mM Tris–HCl pH 7.5, 100 mM NaCl plus 5% glycerol. Protein concentrations were estimated based on $OD_{280}$ and aromatic amino acid content.

Fluorescence labeling of $ParB_{pSM}$ was done as described for Alexa647-ParB$_F$ (*Vecchiarelli et al., 2013*). $ParB_{pSM}$-GCE, in 50 mM Tris–HCl pH 7.4, 100 mM NaCl, 0.1 mM EDTA, 10% glycerol, was mixed with Alexa647 maleimide (Invitrogen) at a protein-to-dye ratio of 1:1 and then incubated for 1 hr at room temperature in the dark. Then, 20 mM DTT was added to stop the reaction. Free dye was removed by spin gel-filtration in a G-50 column. The dye labeling efficiency was determined by spectrophotometry to be ~15%, and the labeled protein was mixed with unlabeled protein to prepare 1%-labeled $ParB_{pSM}$-Alexa647 used for the experiments. In vivo, $ParB_{pSM}$-GCE in combination with $ParA_{pSM}$-GFP was fully competent for partition (Maria Moreno-del Álamo, personal communication), and in vitro, dye labeling did not affect its activities, as measured by its ability to stimulate $ParA_{pSM}$ ATPase and by its $parS_{pSM}$ DNA binding activity (A. Volante, unpublished results).

The N-terminal peptides of $ParB_{pSM}$ (residues 1–27) and $ParB_{P1}$ (residues 1–30) used in the ATPase assays were synthesized by GenScript. The sequence of $ParB_{pSM}^{1-27}$ and its variant $ParB_{pSM}^{1-27\ K10A}$ were NH$_2$-MIVGNLGAQKAKRNDTPISAKKDIMGD-CO$_2$H (≥97% purity) and NH$_2$-MIVGNLGAQAAKRNDTPISAKKDIMGD-CO$_2$H (≥96% purity), respectively. The sequence of $ParB_{P1}^{1-30}$ was NH$_2$-MSKKNRPTIGRTLNPSILSGFDSSSASGDR-CO$_2$H (≥97% purity).

Two strands of each double-stranded DNA containing heptad repeat (5′-WATCACW-3′) or non-consensus scrambled sequences were synthetized, annealed, and purified by ITD (Integrated DNA Technologies). The forward sequences of a DNA duplex are listed in the following table:

| Name | Sequence (5'–3') |
|---|---|
| Scram | 5'-GGG ATCAAAC ACTTGAT AGACAAG TCTTTGA CCTAATT GTGAAAA TTATGAA GGG-3' |
| 7R | 5'-GGG *AATCACA AATCACA AGTGATT AATCACA AATCACT TGTGATT TGTGATT* GGG-3' |
| 6R | 5'-GGG *AATCACA AGTGATT AATCACA AATCACT TGTGATT TGTGATT* GGG-3' |
| 5R | 5'-GGG *AATCACA AATCACA AATCACT TGTGATT TGTGATT* GGG-3' |
| 4R(1) | 5'-GGG *AATCACA AATCACT TGTGATT TGTGATT* GGG-3' |
| 4R(2) | 5'-GGG *AATCACT TATCACA AGTGATT AATCACT* GGG-3' |
| 4R(3) | 5'-GGG *AATCACT TATCACA AATCACA AATCACT* GGG-3' |
| 3R – 2nc – 3R | 5'-GGG *AATCACA AATCACA AATCACA* TCATAGT TCATAGT *TGTGATT TGTGATT TGTGATT* GGG-3' |
| 3R – 1nc – 3R | 5'-GGG *AATCACA AATCACA AATCACA* TCATAGT *TGTGATT TGTGATT TGTGATT* GGG-3' |
| 3R – 1nc | 5'-GGG *AATCACA AATCACT TGTGATT* TCATAGT GGG-3' |
| 3R(1) | 5'-GGG *AATCACA AATCACT TGTGATT* GGG-3' |
| 3R(2) | 5'-GGG *AATCACT TATCACA AATCACA* GGG-3' |
| 2R | 5'-GGG *AATCACT TGTGATT* GGG-3' |
| 1R | 5'-GGG *AATCACT* GGG-3' |

## Plasmids

Plasmids encoding wild-type ParA$_{pSM}$ (Delta, pCB746), mutant ParA$_{pSM}$$^{D60A}$ (pCB755), and ParB$_{pSM}$ (Omega, pT712$\omega$) have been described previously (Welfe et al. 2005, *Pratto et al., 2008*; *Volante and Alonso, 2015*). The pET11a harboring the sequence of his-ParA$_{pSM}$$^{D60E}$ (pET11a-ParA$_{pSM}$$^{D60E}$) his-ParA$_{pSM}$-eGFP (pET11a-ParA$_{pSM}$-eGFP) and its variants (D60A, pET11a-ParA$_{pSM}$$^{D60A}$-eGFP and D60E, pET11a-ParA$_{pSM}$$^{D60E}$-eGFP) were constructed by GenScript. The ParB$_{pSM}$-GCE [72G, 73C, 74E] allele was created by site-directed mutagenesis and then cloned into the ParB$_{pSM}$ expression plasmid (pT712) to generate pCB1033.

## ATPase assays

Unless stated otherwise, all ATPase assays were performed as follows: the reaction contained 2 µM ParA$_{pSM}$, the indicated concentration of full-length ParB$_{pSM}$ (or ParB$_{pSM}$$^{1-27}$), nsDNA (plasmid pBR322 DNA, 60 µM in bp), and *parS$_{pSM}$* DNA duplex in 50 mM Tris–HCl (pH 7.4), 100 mM KCl, 2 mM MgCl$_2$, and 1.5 mM [γ-$^{32}$P]ATP. Labeled ATP was purified before use as previously described (*Vecchiarelli et al., 2010*). Reactions were incubated at 37°C for 3 hr and analyzed by TLC as previously described (*Pratto et al., 2008*). The data points shown are the means with error bars (standard errors of mean) of repeat experiments (N) as indicated for each figure panel. N* indicates repeat number for majority of ParB$_{pSM}$ concentration points as detailed in the Source Data file. Datasets of repeated measurements were fit after subtraction of background measured without ParA$_{pSM}$ to a modified Hill equation: $v - v_0 = (v_{max} [B]^n)/(K_A^n + [B]^n)$, and the fit parameters and symmetrized error ranges reflecting the larger error of the 95% confidence interval below and above the mean were estimated using Prism 9 (GraphPad). For [B], total concentration of ParB$_{pSM}$ was used instead of free ParB$_{pSM}$ concentration due to technical reasons.

## nsDNA-carpeted flow cell preparation

The flow cells coated with lipid bilayer with attached biotin (DOPC plus DOPE-biotin [1%]) were prepared essentially as described in *Han and Mizuuchi, 2010* and rinsed with a buffer containing 25 mM Tris–HCl pH 7.4, 150 mM NaCl, and 5 mM MgCl$_2$ and 0.1 mM CaCl$_2$. Sonicated and biotinylated DNA was prepared as follows: 250 µl of 10 mg/ml sonicated salmon sperm DNA (Sigma) was

sonicated for an additional 5 min (Misonix sonicator 3000, output level 6, pulsed on/off for 10 s each at 16°C) to size-weighted average length of ~500 bp. In order to biotinylate the DNA ends, the sonicated DNA (1 mg/ml) was incubated with 40 µM biotin-17-dCTP (Invitrogen) and 0.6 units TdT (NEB) in the buffer specified by the enzyme manufacturer at 37°C for 30 min. The reaction was stopped by heating at 70°C for 10 min, and unincorporated biotin-17-dCTP was removed by using S-200 HR Microspin columns (GE Healthcare). The DNA was ethanol precipitated and resuspended in TE buffer. To coat the flow cell with sonicated DNA, the DNA prepared as above was dissolved to 1 mg/ml in 25 mM Tris–HCl pH 7.4, 150 mM NaCl, 5 mM MgCl$_2$, and 0.1 mM CaCl$_2$, infused into the assembled flow cell, and incubated overnight at 4°C. Unbound DNA was removed by rinsing with 50 mM Tris–HCl (pH 7.4), 100 mM KCl, 2 mM MgCl$_2$, and 10% glycerol.

## TIRFM setup and image processing

Total internal reflection fluorescence (TIRF) illumination and microscopy, as well as the camera settings, were essentially as described (*Hwang et al., 2013*). Combined beam of a 488 nm diode-pumped, solid-state laser (Coherent) and a 633 nm HeNe laser (Research Electro-Optics) was pointed through a fused silica prism onto the top side of the sample flow cell. Fluorescence emission was collected through a ×40 Plan Apo VC, NA 1.4 oil-immersion objectives (Nikon) and magnifier setting at ×1.5. The laser excitation lines were blocked below objective lens with notch filters (NF03-488E and NF03-633E, Semrock). The fluorescence images were captured by an EMCCD camera (Andor IXON +897) through a Dual-View module (630DCXR cube, Photometrics; short/long pass filters, SP01-633RS, LP02-633RE, Semrock). Microscopy experiments were carried out at room temperature (~23°C). Typical camera settings were digitization 16 bit at 1 MHz, preamplifier gain 5.2, vertical shift speed 2 MHz, vertical clock range: normal, EM gain 40, EMCCD temperature set at –98°C, baseline clamp ON. Images were acquired at exposure time 100 ms with frame rate 1, 0.4, or 0.2 Hz using MetaMorph 7 software (Molecular Devices) and analyzed using ImageJ software (NIH) as described (*Hwang et al., 2013*). Data were analyzed with Prism 9 (GraphPad).

## Estimation of the nsDNA-carpet-bound protein densities from the observed fluorescence intensities

ParA$_{pSM}$ and ParB$_{pSM}$ density on the nsDNA-carpet (dimers/µm$^2$) were calculated from the fluorescence intensity of acquired images essentially according to the procedure described in Figure S4 legend in *Vecchiarelli et al., 2016*. The labeled protein samples used in the experiments were diluted to different concentrations (0–8 µM) in a buffer (50 mM Tris–HCl, pH 7.4, 100 mM KCl, 2 mM MgCl$_2$, 10% glycerol, 1 mM DTT, 1 mg/ml α-casein, and 0.6 mg/ml ascorbic acid) and infused into flow cell coated with 1,2-dioleoyl-sn-glycero-3-phosphocholine. Fluorescence intensity data were collected before the protein arrival (background), with the protein sample in the flow cell, and after washing the flow cell with buffer (background) using the illumination and image acquisition parameter settings used for the series of experiments for which the conversion parameters were prepared. The fluorescence signal from the protein sample in the solution was obtained by subtracting the background signal with buffer only (mostly camera dark noise). From the wavelength, the refractive indices of fused silica slide glass and the reaction buffer, and the illumination angle, the evanescence penetration depths were calculated to be 131 and 170 nm for 488 and 633 nm excitation beams, respectively. The protein concentration and evanescence penetration volume yielded the number of protein molecules, and when bound to the flow cell surface, would produce the observed fluorescence signal, yielding the conversion factor used.

## ParA$_{pSM}$ and ParB$_{pSM}$ association and dissociation from the nsDNA-carpet

Two inlet flow cells were assembled and coated with sonicated salmon sperm DNA as described (*Vecchiarelli et al., 2013*). One inlet was connected to syringe A containing 'binding solution' and the other to syringe B containing 'wash solution.' For protein binding to the nsDNA-carpet, syringe A delivered at 5 µl/min, while syringe B at 0.5 µl/min. To start the wash, the flow rates of the two syringes were reversed keeping the total flow rate constant. Experiments were done at room temperature in pSM19035 buffer: 50 mM Tris–HCl (pH 7.4), 100 mM KCl, 2 mM MgCl$_2$, 10% glycerol, 1 mM DTT, 0.1 mg/ml α-casein, 0.6 mg/ml ascorbic acid, and 1 mM ATP. Additional components of the binding

and washing solutions were as specified in the figure legends. Syringe components were preincubated before infusion at 23°C for 30 min or longer. The association and dissociation data points represent the means with error bars (standard errors of mean) of repeat experiments (N) as indicated for each panel. The dissociation curves were fitted to a single- or a double-exponential equation after subtraction of background in the absence of protein with plateau level set to zero when necessary to avoid large plateau values, and the fraction of fast-dissociating population, the $k_{off,}$ and symmetrized error ranges reflecting the larger error of the 95% confidence interval below and above the mean were estimated using Prism 9 (GraphPad).

## Acknowledgements

We are grateful to our colleagues Keir Neuman, Barbara Funnell, Anthony Vecchiarelli, James Taylor, Michiyo Mizuuchi, Min Li, Masaki Osawa, and William Carlquist for helpful suggestions and discussion, and to Maria Moreno-del Álamo for providing the information concerning the in vivo functionality of ParA$_{pSM}$-GFP/ParB$_{pSM}$-GCE. This work was supported by the intramural research fund for National Institute of Diabetes and Digestive and Kidney Diseases (KM) and the Ministerio de Ciencia e Innovación, Agencia Estatal de Investigación (MCIN/AEI)/FEDER, 2018-097054-B-I00 and 2021AEP031 to JCA.

## Additional information

### Funding

| Funder | Grant reference number | Author |
| --- | --- | --- |
| National Institute of Diabetes and Digestive and Kidney Diseases | | Kiyoshi Mizuuchi |
| Ministerio de Ciencia e Innovación | 2018-097054-B-I00 | Juan Carlos Alonso |
| Ministerio de Ciencia e Innovación | 2021AEP031 | Juan Carlos Alonso |

The funders had no role in study design, data collection and interpretation, or the decision to submit the work for publication.

### Author contributions

Andrea Volante, Conceptualization, Investigation, Visualization, Writing - original draft; Juan Carlos Alonso, Kiyoshi Mizuuchi, Conceptualization, Supervision, Funding acquisition, Writing - review and editing

### Author ORCIDs

Juan Carlos Alonso (ID) http://orcid.org/0000-0002-5178-7179
Kiyoshi Mizuuchi (ID) http://orcid.org/0000-0001-8193-9244

### Decision letter and Author response

Decision letter https://doi.org/10.7554/eLife.79480.sa1
Author response https://doi.org/10.7554/eLife.79480.sa2

## Additional files

### Supplementary files

• MDAR checklist

### Data availability

All data generated or analysed during this study are included in the manuscript and supporting file; source data files have been provided for all figures presented.

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
