## [Editor Report]

The work by Volante et al. studied a plasmid partition system, in which the authors discovered that four or more contiguous ParS sequence repeats are required to assemble a stable partitioning ParAB complex and activate the ParA ATPase. The work reveals a plasmid partitioning mechanism in which the mechanic property of DNA and its interaction with the partition complex may drive the directional movement of the plasmid.

---

## [Decision Letter]

**Decision letter after peer review:**

Thank you for submitting your article "Distinct architectural requirements for the parS centromeric sequence of the pSM19035 plasmid partition machinery" for consideration by *eLife*. Your article has been reviewed by 3 peer reviewers, and the evaluation has been overseen by a Reviewing Editor and Anna Akhmanova as the Senior Editor. The reviewers have opted to remain anonymous.

Essential revisions:

While the reviewers are interested in seeing more following-up in vitro work demonstrating that the torsional stress of DNA is indeed involved in the binding of ParB to four or more contiguous ParS sites, they agree that these additional experiments are unnecessary, given the extensive biochemical characterizations of the system in the manuscript. However, the reviewers would like the authors to discuss or demonstrate the in vivo physiological relevance of the new Par system. This could be done by testing whether or not 4R or 7R is sufficient for plasmid stability. A simple test is to express ParAB from a chromosomally integrated copy under IPTG control and place 4R or 7R on an unstable plasmid and look at retention. Alternatively and at minimum, the authors should include a section in the discussion on how their in vitro finding connects to in vivo conditions.

*Reviewer #1 (Recommendations for the authors):*

Perhaps the only shortcoming of this work is that the team does not yet get to the bottom of WHY four consecutive parS are needed; arguably this would be the most interesting/mechanistic aspect of the pSM19035 ParABS system. The last part of the Discussion (line number 557-582) is highly speculative. Volante et al., proposed: “ We propose the specific MECHANICAL properties of the contiguous ParBpSM-parSpSM mini-filament, likely different from gapped mini-filaments, is required for efficient ATPase activation, explaining the requirement for the parSpSM sequence repeat contiguity”. I wonder if Volante et al., can test, perhaps by AFM/TEM, if four consecutive parS (with ParB bound) induce a special bend/architecture to the DNA? In this manuscript, Volante et al., mentioned that previous work by Weihofen et al., 2006 solved the crystal structure of ParBpSM bound to parS DNA and no bending of the DNA was observed. However, in Weihofen et al., the piece of DNA has only 2 consecutive parS instead of four.

If the model in Figure 6 was true, it would predict that perhaps a (parS-scram-scram-parS) R-2nc-R DNA is as good as a (parS-parS-parS-parS) 4R DNA (because 2 ParB dimers are still on the same helical face to interact with 2 ParA dimers, Figure 6B)? If this (parS-scram-scram-parS) DNA can be included as the control for Figure 1, that perhaps would be revealing/lend more support to the model in Figure 6? Of course, this (parS-scram-scram-parS) DNA can be seen as essentially the same as the 3R-2nc-3R site (Figure 1E) but I think it would be a cleaner control without any possible artifact from 3 consecutive parS on each side.

Lastly, is there cooperativity among DNA-bound ParB that might explain why 4 consecutive sites (but not 3) are optimal?

*Reviewer #2 (Recommendations for the authors):*

I would like to know if there is any information on the ability of a single 7R or 4R to promote plasmid segregation in vivo.

*Reviewer #3 (Recommendations for the authors):*

While mechanistic details of this double-remodeling Brownian Ratchet need to iron out in the future, it is my absolute pleasure to recommend this paper for publication. I only suggest the authors consider to take in the following suggestions:

1. In the introduction, it may be better to frame the work by citing the theoretical models by Hu et al. (Hu et al, PNAS, 2015; Hu et al, Biophysical J, 2017 & 2021). These models assume a simple ParA-ParB binding and suggest that the ParABS-mediated Brownian Ratcheting requires PC/cargo to have enough ParA-ParB bonds for partition fidelity. The authors can then use physical insights to contrast the peculiar case of the pSM19035 plasmid partition system, which would make this work more interesting.

2. For lines 79-108, I suggest restructuring the two paragraphs. Currently, the authors' key finding is mixed with the background, which makes it somewhat difficult to grasp the key point. I'd suggest putting forward the relevant background first and then consolidating the finding.

3. The authors noted that (1) the pSM19035 plasmid partition system probably operates as a Brownian Ratchet, and (2) the ParA here is somewhat different from other systems in that it forms a short, clustered patch, instead of individually distributing on the nucleoid independently. Hypothetically, the clustered ParA patch may still behave like a cytoskeletal filament that may push/pull to drive the directed movement. However, the authors measured the ParA ATPase activity: Its maximal rate is ~ 0.3 ATP per min. This is not only consistent with other ParABS systems (like SopABC) but also too slow to power the directed movement of plasmid/DNA in vivo. I'd suggest the authors elaborate on this point in their draft to put their findings more clearly in the context of Brownian Ratchet, instead of a filament-based pushing/pulling mechanism.

4. Table 1 needs to have units for koff.

5. In the proposed model, the authors elaborated on how the thermal energy can be stored in the forming ParA-ParB bond and then trigger the bond dissociation. But the number of bonds is still just a few and may not be enough to quench the diffusive motion of PC. Is it possible that this special architecture of ParS-ParB increases the effective viscous drag between PC and ParA? Or is it possible that the ParA-ParB bond is much stronger than other systems? It'd be great if the authors could share their views more explicitly in this regard.

---

## [Author Response]

Essential revisions:While the reviewers are interested in seeing more following-up in vitro work demonstrating that the torsional stress of DNA is indeed involved in the binding of ParB to four or more contiguous ParS sites, they agree that these additional experiments are unnecessary, given the extensive biochemical characterizations of the system in the manuscript.

Thank you for the comment. We indeed are very much interested in directly testing our torsional stress hypothesis. However, this would require challenging new experimental design development through testing of several potential approaches, possibly including new single-molecule methods, and we consider this as a subject of our future extension of the project. We added a short sentence at the end of the Discussion mentioning this as one of the requisite tests remaining to be done (line 615-616).

However, the reviewers would like the authors to discuss or demonstrate the in vivo physiological relevance of the new Par system. This could be done by testing whether or not 4R or 7R is sufficient for plasmid stability. A simple test is to express ParAB from a chromosomally integrated copy under IPTG control and place 4R or 7R on an unstable plasmid and look at retention. Alternatively and at minimum, the authors should include a section in the discussion on how their in vitro finding connects to in vivo conditions.

Indeed, we very much wanted to include the in vivo plasmid stability test for the *parS_pSM_* sequence requirement in the current paper. Both the first author of the paper at NIH and a member of the Alonso Lab at Madrid spent significant efforts to set up the stability test system. Their preliminary results initially looked promising, but further test results were confusing, and we suspected the protein expression level control must be improved to obtain reliable data. We also became wary additional aspects of the plasmid structure could also impact the plasmid stability, and more systematic investigation might be required than we planned at this time. With time running out before the first author needed to move on to his new job, we had to give up the in vivo test for this paper. We added a segment in the opening part of the Discussion pointing out that whether the functional requirements for the *parS_pSM_* structure reported here also applies to the in vivo partition activity or not remains to be confirmed, even though we expect the in vivo test to parallel our biochemical results (lines 439-442). We also expanded the discussion concerning the implications of the inferred differences of the reaction parameters of this system from those of HTH-ParB systems on the consideration of the effectiveness as a partition system on lines 581-610, (in part following a suggestion by the reviewer #3).

Reviewer #1 (Recommendations for the authors):Perhaps the only shortcoming of this work is that the team does not yet get to the bottom of WHY four consecutive parS are needed; arguably this would be the most interesting/mechanistic aspect of the pSM19035 ParABS system. The last part of the Discussion (line number 557-582) is highly speculative. Volante et al., proposed: " We propose the specific MECHANICAL properties of the contiguous ParBpSM-parSpSM mini-filament, likely different from gapped mini-filaments, is required for efficient ATPase activation, explaining the requirement for the parSpSM sequence repeat contiguity".

We fully agree this is highly speculative, as we admitted at the start of description of the model on lines 525-526. But this is our current best effort to satisfy the set of curious findings we uncovered in this study, until challenged by further experiments.

I wonder if Volante et al., can test, perhaps by AFM/TEM, if four consecutive parS (with ParB bound) induce a special bend/architecture to the DNA? In this manuscript, Volante et al., mentioned that previous work by Weihofen et al., 2006 solved the crystal structure of ParBpSM bound to parS DNA and no bending of the DNA was observed. However, in Weihofen et al., the piece of DNA has only 2 consecutive parS instead of four.

Thank you for pointing out the hole in the introduction of the prior observation on the *parS*/ParB complex. We revised the introduction on this subject on lines 103-108, clarifying that the straight structure for the seven repeat *parS_pMS_* is a model based on the structures of two-repeat complexes. AFM images of the seven repeat complexes have been published and they support roughly straight configuration of the complex (without noticeable sharp bends).

If the model in Figure 6 was true, it would predict that perhaps a (parS-scram-scram-parS) R-2nc-R DNA is as good as a (parS-parS-parS-parS) 4R DNA (because 2 ParB dimers are still on the same helical face to interact with 2 ParA dimers, Figure 6B)? If this (parS-scram-scram-parS) DNA can be included as the control for Figure 1, that perhaps would be revealing/lend more support to the model in Figure 6? Of course, this (parS-scram-scram-parS) DNA can be seen as essentially the same as the 3R-2nc-3R site (Figure 1E) but I think it would be a cleaner control without any possible artifact from 3 consecutive parS on each side.

Because 3R-2nc-3R contains 1R-2nc-1R as noted by the reviewer, we consider it is unlikely that 1R-2nc-1R exhibit much higher activity, although this is formally possible if the outer 3Rs interfere with (outcompete) the central 1R-2nc-1R part. Since we have not done the experiment, and the first author who did all the experiments is not available to repeat the experiment, we cannot test this possibility in timely fashion at present. In future, when it becomes possible to start testing the “torsional stress” idea, we believe 1R-2nc-1R and 1R-1nc-2R would be useful substrates to be included in the study.

Lastly, is there cooperativity among DNA-bound ParB that might explain why 4 consecutive sites (but not 3) are optimal?

Previous study showed ParB_pSM_ binds with roughly similar apparent Kd (between ~5 to ~15 nM) to 3R and 4R DNA fragments with different orientation combinations, and also to a 10R DNA fragment. We confirmed the 3R and 4R fragments used in this study showed similar affinity to ParB_pSM_ (Kd ~17 nM, Figure 1-Figure sup 3). Gel shift banding patterns indicate binding to the repeated sites are cooperative, although some sign of minor cooperativity difference appears to exist depending on the sequence orientation combinations (de la Hoz et al., 2004). Considering that the concentrations of the protein and DNA fragments used in our experiments are much higher than the apparent Kd of binding, we believe all the *parS_pSM_* fragments with possible exception of R1 are essentially fully occupied by ParB_pSM_ in our experiments. We expanded a few details mentioned above on lines 188-191.

Reviewer #2 (Recommendations for the authors):I would like to know if there is any information on the ability of a single 7R or 4R to promote plasmid segregation in vivo.

We have not studied this question systematically yet, although there are some relevant anecdotal unpublished observations exist. We noted this as another point that remains to be experimentally addressed on lines 442-445.

Reviewer #3 (Recommendations for the authors):While mechanistic details of this double-remodeling Brownian Ratchet need to iron out in the future, it is my absolute pleasure to recommend this paper for publication. I only suggest the authors consider to take in the following suggestions:1. In the introduction, it may be better to frame the work by citing the theoretical models by Hu et al. (Hu et al, PNAS, 2015; Hu et al, Biophysical J, 2017 & 2021). These models assume a simple ParA-ParB binding and suggest that the ParABS-mediated Brownian Ratcheting requires PC/cargo to have enough ParA-ParB bonds for partition fidelity. The authors can then use physical insights to contrast the peculiar case of the pSM19035 plasmid partition system, which would make this work more interesting.

Thank you for the excellent suggestion. We summarized the principal requirements for effective plasmid partition by the diffusion-ratchet concept using HTH-ParB system as the example, and then highlighted potential issues for the system with small number of ParA-ParB bonds at the PC in the introduction at lines 69-85.

2. For lines 79-108, I suggest restructuring the two paragraphs. Currently, the authors' key finding is mixed with the background, which makes it somewhat difficult to grasp the key point. I'd suggest putting forward the relevant background first and then consolidating the finding.

Thank to for the suggestion. We followed the advice combining the parts describing background information into one paragraph and separating it from the short statement on what we did in this study.

3. The authors noted that (1) the pSM19035 plasmid partition system probably operates as a Brownian Ratchet, and (2) the ParA here is somewhat different from other systems in that it forms a short, clustered patch, instead of individually distributing on the nucleoid independently. Hypothetically, the clustered ParA patch may still behave like a cytoskeletal filament that may push/pull to drive the directed movement. However, the authors measured the ParA ATPase activity: Its maximal rate is ~ 0.3 ATP per min. This is not only consistent with other ParABS systems (like SopABC) but also too slow to power the directed movement of plasmid/DNA in vivo. I'd suggest the authors elaborate on this point in their draft to put their findings more clearly in the context of Brownian Ratchet, instead of a filament-based pushing/pulling mechanism.

We followed the suggestion and summarized this point at lines 163-170.

4. Table 1 needs to have units for koff.

We added the unit in the table. Thank you for noticing the mistake.

5. In the proposed model, the authors elaborated on how the thermal energy can be stored in the forming ParA-ParB bond and then trigger the bond dissociation. But the number of bonds is still just a few and may not be enough to quench the diffusive motion of PC. Is it possible that this special architecture of ParS-ParB increases the effective viscous drag between PC and ParA? Or is it possible that the ParA-ParB bond is much stronger than other systems? It'd be great if the authors could share their views more explicitly in this regard.

Thank you again for the helpful suggestion. This goes hand in hand with point #1 above. We added suggested discussion toward the end of the Discussion section at lines 581-610.